# CTCF and cohesin regulate chromatin loop stability with distinct dynamics

Anders S Hansen[1,2,3,4], Iryna Pustova[1,2,3,4], Claudia Cattoglio[1,2,3,4], Robert Tjian[1,2,3,4]*, Xavier Darzacq[1,2,3]*

[1]Department of Molecular and Cell Biology, University of California, Berkeley, Berkeley, United States; [2]Li Ka Shing Center for Biomedical and Health Sciences, University of California, Berkeley, Berkeley, United States; [3]CIRM Center of Excellence, University of California, Berkeley, Berkeley, United States; [4]Howard Hughes Medical Institute, University of California, Berkeley, Berkeley, United States

**Abstract** Folding of mammalian genomes into spatial domains is critical for gene regulation. The insulator protein CTCF and cohesin control domain location by folding domains into loop structures, which are widely thought to be stable. Combining genomic and biochemical approaches we show that CTCF and cohesin co-occupy the same sites and physically interact as a biochemically stable complex. However, using single-molecule imaging we find that CTCF binds chromatin much more dynamically than cohesin (~1–2 min vs. ~22 min residence time). Moreover, after unbinding, CTCF quickly rebinds another cognate site unlike cohesin for which the search process is long (~1 min vs. ~33 min). Thus, CTCF and cohesin form a rapidly exchanging 'dynamic complex' rather than a typical stable complex. Since CTCF and cohesin are required for loop domain formation, our results suggest that chromatin loops are dynamic and frequently break and reform throughout the cell cycle.

*For correspondence: jmlim@ berkeley.edu (RT); darzacq@ berkeley.edu (XD)

## Introduction

Mammalian interphase genomes are functionally compartmentalized into topologically associating domains (TADs) spanning hundreds of kilobases. TADs are defined by frequent chromatin interactions within themselves and they are insulated from adjacent TADs (*Dekker and Mirny, 2016*; *Dixon et al., 2012*; *Hu et al., 2015*; *Merkenschlager and Nora, 2016*; *Nora et al., 2012*; *Wang et al., 2016*). Most TAD or domain boundaries are strongly enriched for CTCF (*Figure 1A*), an 11-zinc finger DNA-binding protein (*Ghirlando and Felsenfeld, 2016*), and cohesin (*Figure 1B*), a ring-shaped multi-protein complex composed of Smc1, Smc3, Rad21 and SA1/2 that is thought to topologically entrap DNA (*Ivanov and Nasmyth, 2005*; *Skibbens, 2016*). The subset of TADs which are folded into loops are referred to as loop domains and tend to be demarcated by convergent CTCF-binding sites (*Rao et al., 2014*). Targeted deletions of CTCF-binding sites demonstrate that CTCF causally determines loop domain boundaries (*Guo et al., 2015*; *Sanborn et al., 2015*; *de Wit et al., 2015*). Moreover, disruption of loop domain boundaries by deletion or silencing of CTCF-binding sites allows abnormal contact between previously separated enhancers and promoters, which can induce aberrant gene activation leading to cancer (*Flavahan et al., 2016*; *Hnisz et al., 2016a*) or developmental defects (*Lupiáñez et al., 2015*). Finally, genetically engineered depletion of both CTCF (*Nora et al., 2017*) and cohesin (*Schwarzer et al., 2016*) causes most loops to disappear. Yet, despite much progress in characterizing TADs and loop domains, how they are formed and maintained remains unclear. Since CTCF and cohesin causally control domain organization, here we investigated their dynamics and nuclear organization using single-molecule imaging in live cells.

**eLife digest** A human cell contains about 2 meters of DNA tightly packed in a compartment called the nucleus. Within the space inside the nucleus, different parts of the DNA fold into distinct bundles known as domains. These domains are important for organising the genome and are crucial for regulating gene expression, by stimulating specific DNA segments to activate certain genes. Previous research has shown that DNA segments within the same domain frequently interact, whereas DNA segments in different domains rarely do.

The domains are often folded into loops that are held together by a ring-shaped protein complex called cohesin, while another protein called CTCF positions cohesin and thereby sets the boundaries between the domains. Some mutations are known to disrupt these boundaries, which allows certain DNA segments to activate the wrong genes. This can lead to cancer or cause defects when embryos are developing. However, we do not currently understand how these domains are formed or maintained. In particular, it was unclear whether these loop domains are stable or dynamic structures.

Hansen et al. addressed these questions in embryonic stem cells from mice and human cancer cells. It was found that cohesin and CTCF form a complex that binds to the DNA and likely holds the loops together. In further experiments, single molecules of cohesin and CTCF were tracked inside cells using super-resolution microscopy. The results showed that CTCF and cohesin bind to DNA with different dynamics: CTCF binds the DNA for about a minute, whereas cohesin binds the DNA for about 20–25 minutes. Once CTCF detaches from DNA, it quickly rebinds DNA at another site, but cohesin takes much longer. These observations suggest that rather than remaining static, chromatin domains are held together by a dynamic protein complex, with a molecular composition that exchanges over time. This results suggests that DNA loop domains, which were generally assumed to be very stable anchor points, are in fact highly dynamic structures that frequently fall apart and reform.

The next challenge will be to understand how the dynamic nature of these loop domains contribute to gene regulation. This may, one day, enable us to manipulate the domains to correct faulty folding of DNA in cancer and other diseases.

## Results

### CTCF and cohesin form a loop maintenance complex

In order to image CTCF and cohesin without altering their endogenous expression levels, we used CRISPR/Cas9-mediated genome editing to homozygously tag *Ctcf* and *Rad21* with HaloTag in mouse embryonic stem (mES) cells (*Figure 1C*, clones C87 and C45). We also generated a double Halo-mCTCF/mRad21-SNAP$_f$ knock-in mESC line (*Figure 1C*, C59) as well as a Halo-hCTCF knock-in human U2OS cell line (*Figure 1C*, C32). Halo- and SNAP$_f$-Tags can be covalently conjugated with bright cell-permeable small molecule dyes suitable for single-molecule imaging (*Figure 1D*; *Figure 1—figure supplement 1*; *Grimm et al., 2015*). To examine the effect of tagging CTCF and Rad21, which are both essential proteins, we performed control experiments in the doubly tagged mESC line (C59), and observed no effect on mESC pluripotency in a teratoma assay (*Figure 1—figure supplement 2*), expression of key stem cell genes (*Figure 1—figure supplement 3A*) or tagged protein abundance (*Figure 1—figure supplement 3B*). Next, to further validate our endogenous tagging approach, we performed chromatin immunoprecipitation followed by DNA sequencing (ChIP-Seq) using antibodies against CTCF and Rad21 in both wild-type (wt) and the double knock-in C59 line. We compared ChIP-Seq enrichment for both wt and C59 at called wt peaks and observed similar enrichment (*Figure 1E–F*). Notably, 97% of the 33,434 called Rad21 peaks co-localize with one of the 68,077 called CTCF peaks (*Figure 1—figure supplements 4–5*; *Supplementary file 1*), suggesting an intrinsic link between CTCF and cohesin and largely confirming previous reports of ~70–90% overlap (*Parelho et al., 2008*; *Wendt et al., 2008*). However, chromatin co-occupancy by ChIP-seq at the same sites does not necessarily mean that CTCF and Rad21 bind simultaneously. Thus, to determine whether CTCF and cohesin physically interact, we performed co-

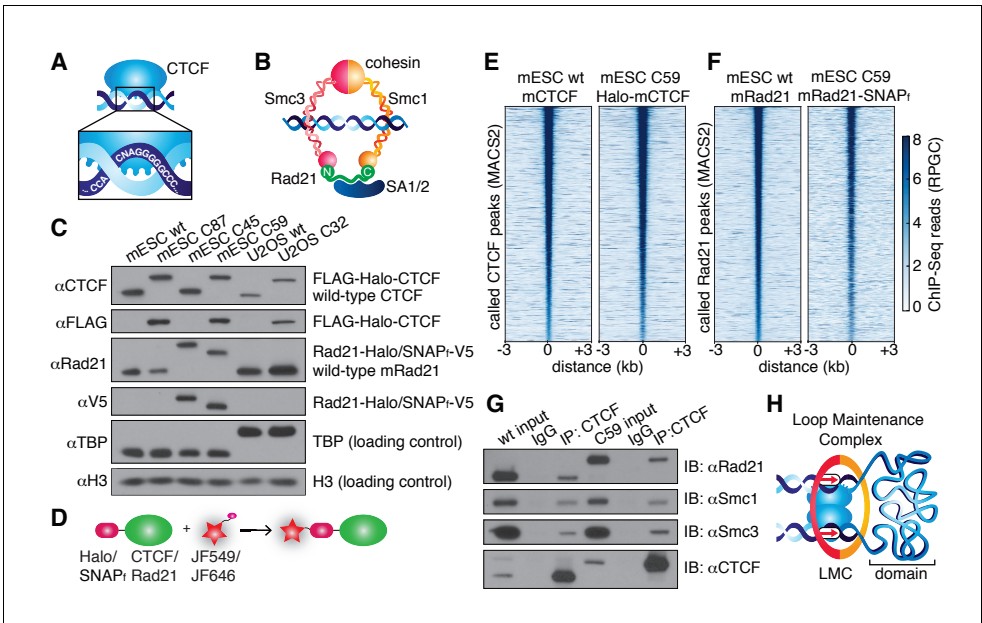

**Figure 1.** CTCF and cohesin can be endogenously tagged and form a complex. (**A**) Sketch of CTCF and its consensus DNA-binding sequence. (**B**) Sketch of cohesin, with subunits labeled, topologically entrapping DNA. (**C**) Western blot of mESC and U2OS wild-type (wt) and knock-in cell lines demonstrating homozygous insertions. (**D**) Sketch of covalent dye-conjugation for Halo or SNAP$_f$-Tag. (**E**) CTCF ChIP-Seq read count (Reads Per Genomic Content) for wild-type and C59 plotted at MAC2-called wt-CTCF peak regions centered around the peak. (**F**) Rad21 ChIP-Seq read count (Reads Per Genomic Content) for wild-type and C59 plotted at MACS2-called wt-Rad21 peak regions. (**G**) Co-IP. CTCF was immunoprecipitated and we immunoblotted for cohesin subunits Rad21, Smc1 and Smc3. (**H**) Sketch of a loop maintenance complex (LMC) composed of CTCF and cohesin holding together a spatial domain as a loop.

The following figure supplements are available for figure 1:

**Figure supplement 1.** Specific labeling of HaloTagged and SNAP$_f$-Tagged proteins in live cells.

**Figure supplement 2.** Teratoma assay demonstrates that tagging CTCF and Rad21 does not affect pluripotency in mESCs.

**Figure supplement 3.** Tagging CTCF and Rad21 does not affect expression of key pluripotency genes or CTCF and Rad21 protein levels.

**Figure supplement 4.** CTCF and Rad21 ChIP-Seq results in wt and C59 mESCs.

**Figure supplement 5.** Tagging CTCF and Rad21 does not affect the ChIP-Seq genomic binding pattern.

---

immunoprecipitation (co-IP) studies. CTCF IP pulled down cohesin subunits Rad21, Smc1 and Smc3 in both wt and C59 mES cells (*Figure 1G*), demonstrating a physical interaction between CTCF and cohesin, which is not affected by endogenous tagging.

Together, our ChIP-Seq co-localization (97% of Rad21 peaks overlap with a CTCF peak) and co-IP interaction studies suggest that CTCF and cohesin form a complex on chromatin. The Hi-C study with the highest resolution found ~10,000 loops in human GM12878 cells using very conservative and stringent loop calling and found these loops to be largely conserved between cell types and between mouse and human (*Rao et al., 2014*). Since each loop is anchored by at least two CTCF/cohesin ChIP-Seq-called sites, but often by clusters of CTCF/cohesin sites, we estimate (see Appendix 1 for a full discussion) that at least one-third of cognate-bound CTCF molecules and the majority of chromatin-bound G1 cohesin molecules are involved in chromatin looping. Integrating these

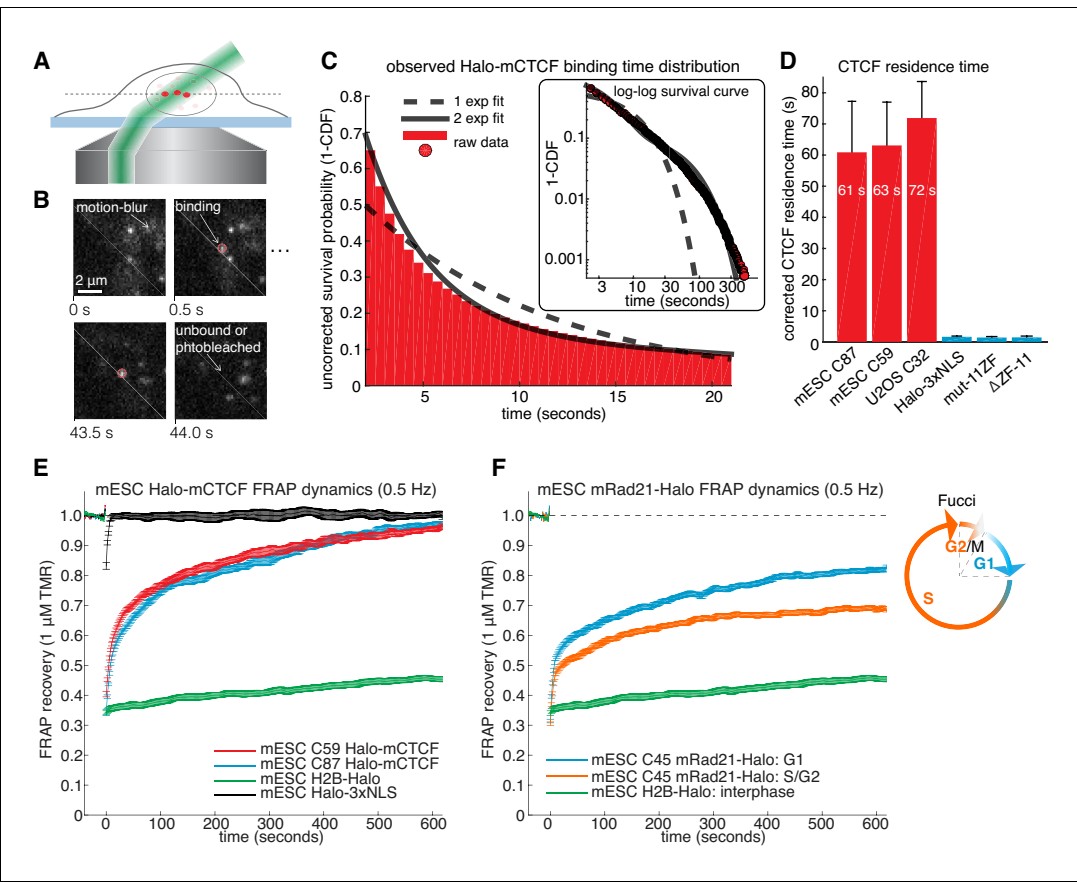

**Figure 2.** CTCF and cohesin have very different residence times on chromatin. (A) Sketch illustrating HiLo (highly inclined and laminated optical sheet illumination) (*Tokunaga et al., 2008*). (B) Example images showing single Halo-mCTCF molecules labeled with JF549 binding chromatin in a live mES cell. (C) A plot of the uncorrected survival probability of single Halo-mCTCF molecules and one- and two-exponential fits. Right inset: a log-log survival curve. (D) Photobleaching-corrected residence times for Halo-CTCF, Halo-3xNLS and a zinc-finger (11 His→Arg point-mutations) mutant or entire deletion of the zinc-finger domain. Error bars show standard deviation between replicates. For each replicate, we recorded movies from ~6 cells and calculated the average residence time using H2B-Halo for photobleaching correction. Each movie lasted 20 min with continuous low-intensity 561 nm excitation and 500 ms camera integration time. Cells were labeled with 1–100 pM JF549. (E) FRAP recovery curves for Halo-mCTCF, H2B-Halo and Halo-3xNLS in mES cells labeled with 1 μM Halo-TMR. (F) FRAP recovery curves for mRad21-Halo and H2B-Halo in mES cells labeled with 1 μM Halo-TMR. Right: sketch of Fucci cell-cycle phase reporter (*Sakaue-Sawano et al., 2008*; *Sladitschek and Neveu, 2015*). We modified the system to contain mCitrine-hGem(aa1-110) and SCFP3A-hCdt(aa30-120) to avoid overlap in the red region of the electromagnetic spectrum. Each FRAP curve shows mean recovery from >15 cells from ≥3 replicates and error bars show the standard error.

The following figure supplements are available for figure 2:

**Figure supplement 1.** Illustration of how residence times are inferred from SMT and control experiments.

**Figure supplement 2.** Supplementary and control CTCF FRAP experiments.

**Figure supplement 3.** Supplementary and control cohesin FRAP experiments.

**Figure supplement 4.** Validation of Fucci reporters.

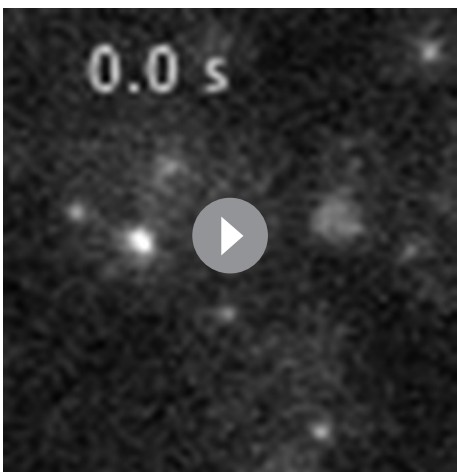

**Video 1.** Single-molecule tracking of Halo-mCTCF in mESCs at 2 Hz. Related to *Figure 2*. Using long 500 ms camera integration causes most diffusing molecules to 'motion-blur' into the background. Laser: 561 nm. Dye: JF549. One pixel: 160 nm.

results with the recent demonstrations (*Nora et al., 2017*; *Schwarzer et al., 2016*) that CTCF and cohesin are causally required for chromatin looping, we refer to the subpopulation of CTCF and cohesin involved in looping as a 'loop maintenance complex' (LMC; *Figure 1H*).

## CTCF and cohesin bind chromatin with very different dynamics

To investigate the dynamics of the LMC, we measured the residence time of CTCF and cohesin on chromatin. First, we used highly inclined and laminated optical sheet illumination (*Tokunaga et al., 2008*) (*Figure 2A*) and single-molecule tracking (SMT) to follow single Halo-CTCF molecules in live cells. By using long exposure times (500 ms), to 'motion-blur' fast moving molecules into the background (*Chen et al., 2014*), we could visualize and track individual stable CTCF-binding events (*Figure 2B*; *Video 1*). We recorded thousands of binding event trajectories and calculated their survival probability. A double-exponential function, corresponding to specific and non-specific DNA binding (*Chen et al., 2014*), was necessary to fit the Halo-CTCF survival curve (*Figure 2C*). After correcting for photo-bleaching (*Figure 2—figure supplement 1A*), we estimated an average residence time (RT) of ~1 min for CTCF in mES cells and a slightly longer RT in U2OS cells (*Figure 2D*). DNA-binding defective CTCF mutants or Halo-3xNLS alone interacted very transiently with chromatin (RT ~1 s; *Figure 2D*). The measured RT did not depend on the dye or exposure time (*Figure 2—figure supplement 1B*). We note that a CTCF RT of ~1 min is a genomic average and that some binding sites likely exhibit a slightly longer or shorter mean residence time. We also note that there is likely an oversampling of binding events at CTCF-binding sites showing the strongest ChIP-Seq enrichment (*Figure 1E*), which tend to be the sites involved in looping (*Merkenschlager and Nora, 2016*). To cross-validate these results using an orthogonal technique, we performed fluorescence recovery after photo-bleaching (FRAP) on Halo-CTCF and quantified the dynamics of recovery (*Figure 2—figure supplement 2A–B*). Both Halo-CTCF in mES cells (*Figure 2E*) and Halo-hCTCF in U2OS cells

**Table 1.** Nuclear search mechanism parameters. *Table 1* lists key parameters for the nuclear search mechanism inferred from model fitting of the displacements in *Figure 3* and the residence times in *Figure 2*.

| | Fraction bound (specific) | Fraction bound (nonspecific) | Free 3D diffusion fraction | Apparent $D_{FREE}$ ($\mu m^2/s$) | $\tau_{SEARCH}$ (total) | Fraction of $\tau_{SEARCH}$ in free 3D diffusion | Fraction of $\tau_{SEARCH}$ in non-specific chromatin association |
|---|---|---|---|---|---|---|---|
| mESC C59 Halo-mCTCF | 48.9% | 19.1% | 32.0% | 2.5 | 65.9 s | 41.3 s | 24.6 s |
| mESC C87 Halo-mCTCF | 49.3% | 19.1% | 31.6% | 2.3 | 62.6 s | 39.0 s | 23.6 s |
| U2OS C32 Halo-hCTCF | 39.8% | 17.7% | 42.5% | 2.5 | 102.8 s | 71.9 s | 30.9 s |
| mESC C45 mRad21-Halo: G1 | 39.8% | 13.7% | 46.5% | 1.5 | 33.0 min | 25.5 min | 7.5 min |
| mESC C45 mRad21-Halo: S/G2 | 49.8% | 13.7% | 36.5% | 1.5 | n/a | n/a | n/a |

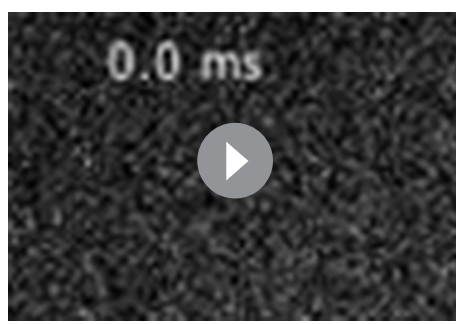

**Video 2.** Single-molecule tracking of Halo-mCTCF in mESCs at 225 Hz. Related to *Figure 3*. Stroboscopic (1 ms of 633 nm) paSMT allows tracking of fast-diffusing molecules. Lasers: 405 and 633 nm. Dye: PA-JF646. One pixel: 160 nm.

(*Figure 2—figure supplement 2C*) exhibited FRAP recoveries consistent with a RT ~1 min, but fitting the FRAP curves with a reaction-dominant model suggested a RT of 3–4 min (*Figure 2—figure supplement 2D*). Whereas our SMT measurements are limited by photobleaching, estimating RTs from FRAP modeling is more indirect and tends to significantly overestimate the RT of transcription factors (*Mazza et al., 2012*) and is also affected by anomalous diffusion. Therefore, we interpret 1 min as a lower bound and 4 min as an upper bound for CTCF's RT in mESCs, but expect the true RT to be closer to 1 min than 4 min.

Our results differ considerably from a previous CTCF FRAP study using over-expressed transgenes, which reported rapid 80% recovery in 20 s (*Nakahashi et al., 2013*). However, when we used similar transiently over-expressed Halo-CTCF instead of endogenous knock-in cells, we also observed similarly rapid recovery (*Figure 2—figure supplement 2B*), suggesting that over-expression of target proteins can result in artefactual measurements. This finding underscores the importance of studying endogenously tagged and functional proteins. Thus, although CTCF (RT ~1–2 min) binds chromatin much more stably than most sequence-specific transcription factors (RT ~2–15 s) (*Chen et al., 2014*; *Mazza et al., 2012*), its binding is still highly dynamic.

We next investigated the cell-cycle dependent cohesin binding dynamics (*Gerlich et al., 2006*). In addition to its role in holding together chromatin loops, cohesin mediates sister chromatid cohesion from replication in S-phase to mitosis. Thus, since TAD demarcation is strongest in G1 before S-phase (*Naumova et al., 2013*), we reasoned that cohesin dynamics in G1 should predominantly reflect the chromatin looping function of cohesin. To control for the cell-cycle, we deployed the Fucci system (*Sakaue-Sawano et al., 2008*) to distinguish G1 from S/G2-phase using fluorescent reporters in the C45 and C59 mESC lines (*Figure 2—figure supplements 3A* and *4*). We then performed FRAP on mRad21-Halo (*Figure 2F*) and mRad21-SNAP_f (*Figure 2—figure supplement 3B*). We observed significantly faster mRad21 recovery in G1 than in S/G2-phase consistent with

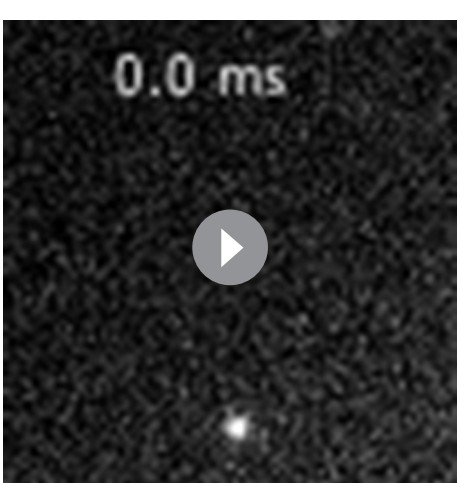

**Video 3.** Single-molecule tracking of ΔZF-Halo-mCTCF in transiently transfected mESCs at 225 Hz. Related to *Figure 3*. Stroboscopic (1 ms of 633 nm) paSMT allows tracking of fast-diffusing molecules. Lasers: 405 and 633 nm. Dye: PA-JF646. One pixel: 160 nm.

*Gerlich et al. (2006)*, but nevertheless much slower recovery than CTCF and CTCF showed the same recovery in G1 and S/G2 (*Figure 2—figure supplement 2E*). The slow mRad21 turnover precluded SMT experiments. Model-fitting of the G1 mRad21 FRAP curves (*Figure 2—figure supplement 3C*) revealed an RT ~22 min. Previous cohesin FRAP studies have reported differing RTs (*Gerlich et al., 2006*; *Huis in 't Veld et al., 2014*) and as was seen for CTCF, over-expressed mRad21-Halo also showed much faster recovery than endogenous mRad21-Halo (*Figure 2—figure supplement 3D*). Although we cannot completely exclude a very small population (<5%) of CTCF or cohesin molecules with a somewhat shorter or longer RT, these RTs reflect chromatin-bound CTCF/cohesin. Since at least one-third of CTCF and the majority of G1 cohesin molecules bound to chromatin mediate looping (see Appendix 1 for estimate), we are confident that these RTs hold for most CTCF/cohesin molecules involved in looping.

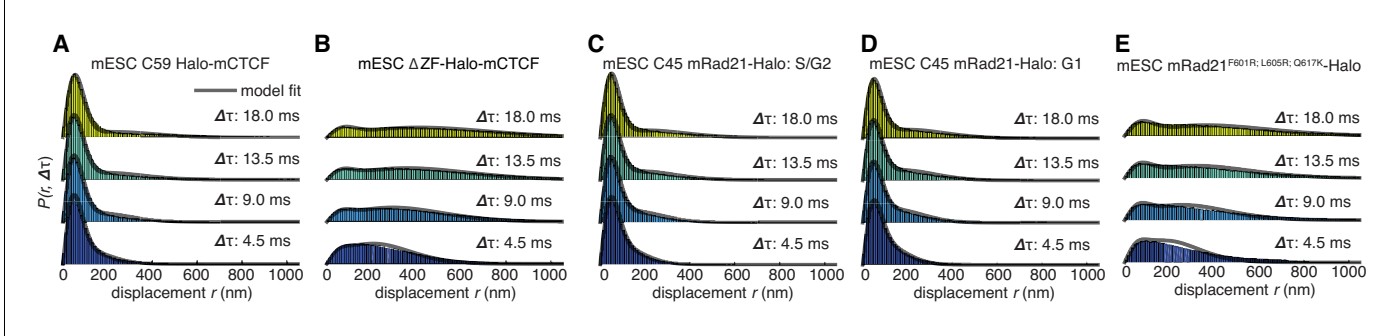

**Figure 3.** Dynamics of CTCF and cohesin's nuclear search mechanism. Single-molecule displacements from ~225 Hz stroboscopic (single 1 ms 633 nm laser pulse per camera integration event) paSMT experiments over multiple time scales for (**A**) C59 Halo-mCTCF, (**B**) a Halo-mCTCF mutant with the zinc-finger domain deleted, C45 mRad21-Halo in S/G2 phase (**C**) and G1 phase (**D**) and (**E**) a Rad21 mutant that cannot form cohesin complexes. Kinetic model fits (three fitted parameters) to raw displacement histograms are shown as black lines. All calculated and fitted parameters are listed in *Table 1*. Displacement histograms were obtained by merging data from at least 24 cells from at least three replicates.

The following figure supplement is available for figure 3:

**Figure supplement 1.** Supplementary stroboscopic paSMT experiments and controls.

Overall, while kinetic modeling of FRAP curves should be interpreted with some caution (*Mazza et al., 2012*), these results, nevertheless, demonstrate a surprisingly large (~10–20x) difference in RTs between CTCF and cohesin, which is difficult to reconcile with the notion of a biochemically stable LMC assembled on chromatin. However, although CTCF and cohesin do not form a stable complex on chromatin, it is still possible that CTCF and cohesin form a stable complex in solution when not bound to DNA.

## CTCF and cohesin exhibit distinct nuclear search mechanisms

To investigate this possibility, we analyzed how CTCF and cohesin each explore the nucleus. Tracking fast-diffusing molecules has been a major challenge. To overcome this issue, we took advantage of bright new dyes (*Grimm et al., 2016*) and developed stroboscopic (*Elf et al., 2007*) photo-activation (*Manley et al., 2008*) single-molecule tracking (paSMT; *Figure 3—figure supplement 1A*), which makes tracking unambiguous (Materials and methods). We tracked individual Halo-mCTCF molecules at ~225 Hz and plotted the displacements between frames (*Figure 3A*). Most Halo-mCTCF molecules exhibited displacements similar to our localization error (~35 nm; Materials and methods) indicating chromatin association, whereas a DNA-binding defective CTCF mutant exhibited primarily long displacements consistent with free diffusion (*Figure 3B*; *Videos 2–3*). To characterize the nuclear search mechanism, we performed kinetic modeling of the measured displacements (*Figure 3—figure supplement 1B*; Materials and methods; *Mazza et al., 2012*) and found that in mES cells, ~49% of CTCF is bound to cognate sites, ~19% is non-specifically associated with chromatin (e.g. 1D sliding or hopping) and ~32% is in free 3D diffusion (*Table 1*). Thus, after dissociation from a cognate site, CTCF searches for ~66 s on average before binding the next cognate site: ~65% of the total nuclear search is random 3D diffusion (~41 s on average), whereas ~35% (~25 s on average) consists of intermittent non-specific chromatin association (e.g. 1D sliding; *Table 1*; note this search time is based on a CTCF RT of ~1 min). The nuclear search mechanism of CTCF in human U2OS cells was similar albeit slightly less efficient (*Table 1*; *Figure 3—figure supplement 1F*). We note that CTCF's search mechanism, with similar amounts of 3D diffusion and 1D sliding, is close to optimal according to the theory of facilitated diffusion (*Mirny et al., 2009*).

Similar analysis of mRad21-Halo in G1 and S/G2 (*Figure 3C–D*) revealed that cohesin complexes diffuse rather slowly compared to CTCF (*Table 1*) and that roughly half of cohesins are topologically engaged with chromatin (G1: ~40%; S/G2: ~50%) compared to ~13% in non-specific, non-topological chromatin association and the remainder in 3D diffusion (G1: ~47%; S/G2: ~37%). Conversely, a Rad21 mutant (*Haering et al., 2004*) unable to form cohesin complexes displayed rapid diffusion and little chromatin association (*Figure 3E*). Like this Rad21 mutant, overexpressed wild-type

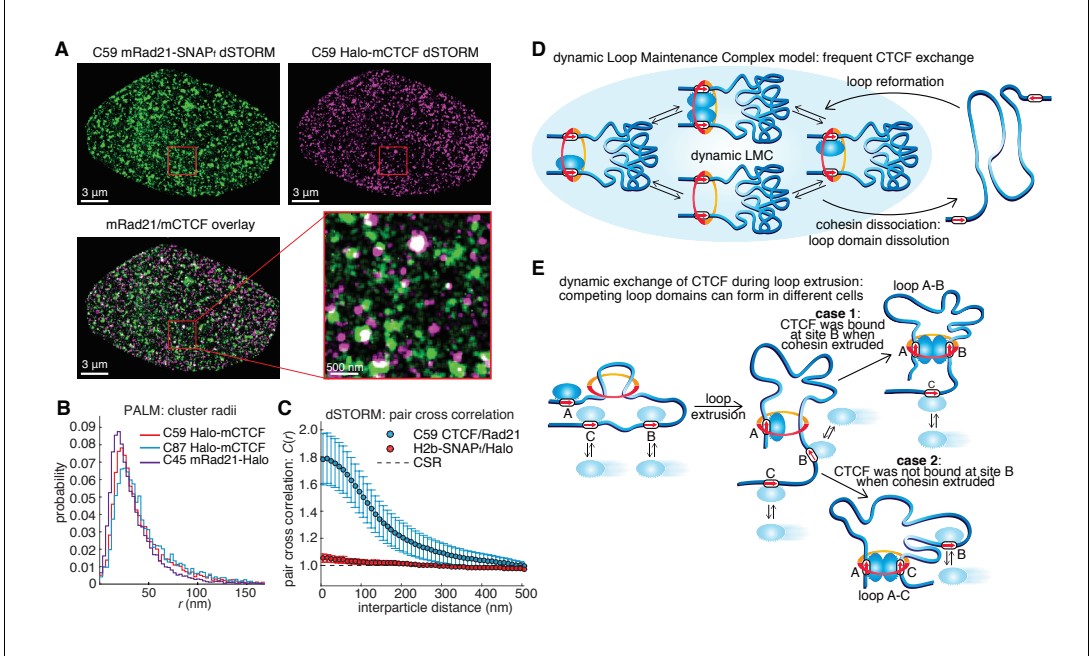

**Figure 4.** Models of CTCF/cohesin mediated chromatin loop dynamics. (A) Two-color dSTORM of C59 mESCs with mRad21-SNAP$_f$ labeled with 500 nM JF549 (green) and Halo-mCTCF labeled with 500 nM JF646 (magenta). High-intensity co-localization is shown as white. Low-intensity co-localization is not visible. Zoom-in on red 3 μm square. Note, the SNAP dye cp-JF549 shows slight artefactual labeling of the nuclear envelope, which was removed during image rendering. (B) Cluster radii distributions for CTCF (C87 and C59) and Rad21 (C45) from single-color PALM experiments using PA-JF549 dyes. (C) Pair cross correlation of C59 and mESC H2B-SNAP$_f$ co-expressing Halo-only. Error bars are standard error from 12 to 18 dSTORM-imaged cells over three replicates. (D) Sketch illustrating the concept of a dynamic loop maintenance complex (LMC) composed of CTCF and cohesin with frequent CTCF exchange and slow, rare cohesin dissociation, which causes loop deformation and topological re-orientation of chromatin. (E) Sketch illustrating how dynamic CTCF exchange during loop extrusion of cohesin may explain alternative loop formations when two competing convergent sites (B and C) for another site A) exist.

The following figure supplement is available for figure 4:

**Figure supplement 1.** Overview of super-resolution PALM approach and control experiments.

mRad21-Halo also showed negligible chromatin association (*Figure 3—figure supplement 1E*) again underscoring the importance of studying endogenously tagged proteins at physiological concentrations. Importantly, this also shows that essentially all endogenously expressed mRad21-Halo proteins are incorporated into cohesin complexes. Topological association and dissociation of cohesin is regulated by a complex interplay of co-factors such as Nipbl, Sororin and Wapl (*Skibbens, 2016*). If we, nevertheless, apply a simple two-state model to analyze cohesin dynamics (Materials and methods), we estimate an average search time of ~33 min in between topological engagements of chromatin in G1, with ~77% of the total search time spent in 3D diffusion (~26 min) compared to ~23% in non-specific chromatin association (7 min). Thus, for each specific topological cohesin chromatin binding-unbinding cycle in G1, CTCF binds and unbinds its cognate sites ~20–30 times. These results are certainly not consistent with a model wherein CTCF and cohesin form a stable LMC. Moreover, since CTCF diffuses much faster than cohesin (*Table 1*), it also seems unlikely that CTCF and cohesin form stable complexes in solution.

## CTCF and cohesin co-localize in cells and show a clustered nuclear organization

To resolve these apparently paradoxical findings, we investigated the nuclear organization of CTCF and cohesin simultaneously in the same nucleus. We labeled Halo-mCTCF and mRad21-SNAP$_f$ in C59 mES cells with the spectrally distinct dyes JF646 and JF549 (*Grimm et al., 2015*), respectively, and performed two-color direct stochastic optical reconstruction microscopy (dSTORM) super-

resolution imaging in formaldehyde-fixed cells (*Figure 4A*). We localized individual CTCF and Rad21 molecules with a precision of ~20 nm, less than half the size of the cohesin ring. We observe significant clustering of both CTCF and Rad21 and a large fraction of these clusters overlap (*Figure 4A* and *Figure 4—figure supplement 1A–C*). We next confirmed clustering using photo-activation localization microscopy (PALM) and found that both CTCF and Rad21 predominantly form small clusters (*Figure 4B* and *Figure 4—figure supplement 1*; mean cluster radius ~30–40 nm). To determine whether individual CTCF and cohesin molecules co-localize, we calculated the pair cross correlation, $C(r)$ (*Stone and Veatch, 2015*). $C(r)$ quantifies spatial co-dependence as a function of length, $r$, and $C(r)=1$ for all $r$ under complete spatial randomness (CSR). CTCF and cohesin exhibited significant co-localization ($C(r)>1$) at very short distances in mES cells (*Figure 4C*). Conversely, CTCF and cohesin were nearly independent at length scales beyond the diffraction limit, emphasizing the importance of super-resolution approaches. A mES cell line co-expressing histone H2B-SNAP$_f$ and Halo proteins imaged under the same dSTORM conditions showed no pair cross-correlation (*Figure 4C*), thereby ruling out technical artifacts. Thus, our two-color dSTORM results provide compelling evidence that a large fraction of CTCF and cohesin molecules indeed co-localize at the single-molecule level inside the nucleus consistent with the LMC model and reveals a clustered nuclear organization.

## Discussion

Chromatin loop domains are widely believed to be very stable structures (*Andrey et al., 2017*; *Ghirlando and Felsenfeld, 2016*; *Hnisz et al., 2016b*) held together by a LMC composed of two CTCFs and cohesin (whether cohesin acts as a single ring or as a pair of rings remains a matter of debate [*Skibbens, 2016*]). While our in vitro biochemical (*Figure 1G*) and co-localization (*Figure 4A–C*) experiments do demonstrate complex formation between CTCF and cohesin, our SMT experiments paradoxically reveal this complex to be highly transient and dynamic (*Figures 2–3*). To reconcile these observations, we therefore propose a 'dynamic LMC' model. Consistent with previous studies, CTCF mainly functions to position cohesin at loop boundaries, whereas cohesin physically holds together the two chromatin strands. However, in the 'dynamic LMC' model, while cohesin holds together a given chromatin loop, different CTCF molecules are frequently alighting and departing in a dynamic exchange thus giving rise to a 'transient protein complex' with a molecular stoichiometry that cycles over time (*Figure 4D*). Since topological chromatin association of cohesin is infrequent (~33 min in G1), dissociation of cohesin (~22 min) likely causes the loop to fall apart (*Figure 4D*). Even if the CTCF and cohesin co-clusters that we observe (*Figure 4A–C*; *Figure 4—figure supplement 1*) are LMC clusters that hold together loop domains, their lifetimes are unlikely to be more than 1–2 hr. Thus, our results suggest that chromatin loops are continuously formed and dissolved throughout a typical 14–24 hr mammalian cell cycle.

Our results suggesting that loops are dynamic also provide experimental support for theoretical polymer simulation studies, which found that only dynamic, but not static, loop structures can reproduce experimentally observed chromatin interaction frequencies (*Benedetti et al., 2014*; *Fudenberg et al., 2016*; *Giorgetti et al., 2014*; *Sanborn et al., 2015*). We note that our quantitative characterization of CTCF and cohesin dynamics could be useful for parameterizing future polymer models. While our results indicate that loops are highly dynamic, the question of how they are formed remains. An attractive but not yet verified recent model suggests that loops are formed by cohesin-mediated loop extrusion (*Fudenberg et al., 2016*; *Sanborn et al., 2015*), whereby cohesin extrudes a loop by sliding on DNA (*Davidson et al., 2016*; *Lengronne et al., 2004*; *Nasmyth, 2001*; *Stigler et al., 2016*) until it encounters two convergent and bound CTCF sites (*Figure 4E*). Our imaging experiments (*Figures 2–3*) cannot readily distinguish cohesin stably bound at loop anchors from cohesin in the process of extrusion and thus our measured residence time of ~22 min reflects the average total duration of both. In the context of the loop extrusion model, our results suggest a mechanism for boundary permeability through dynamic and stochastic CTCF occupancy at cognate CTCF sites, which may explain the formation of competing loop domains (*Figure 4E*). This would also explain why DNA-FISH measurements show that most loops are only present in a subset of cells at any given time (*Sanborn et al., 2015*; *Williamson et al., 2014*). Finally, the highly dynamic view of frequently breaking and forming chromatin loops presented here may also facilitate dynamic long-distance enhancer-promoter scanning of DNA in cis, which may be important for temporally efficient regulation of gene expression.

## Materials and methods

### Cell culture, stable cell line construction and dye labeling

JM8.N4 mouse embryonic stem cells (*Pettitt et al., 2009*) (Research Resource Identifier: RRID: CVCL_J962; obtained from the KOMP Repository at UC Davis) were grown on plates pre-coated with a 0.1% autoclaved gelatin solution (Sigma-Aldrich, St. Louis, MO, G9391) under feeder-free condition in knock-out DMEM with 15% FBS and LIF (full recipe: 500 mL knockout DMEM (ThermoFisher, Waltham, MA, #10829018), 6 mL MEM NEAA (ThermoFisher #11140050), 6 mL GlutaMax (ThermoFisher #35050061), 5 mL Penicillin-streptomycin (ThermoFisher #15140122), 4.6 µL 2-mercapoethanol (Sigma-Aldrich M3148), 90 mL fetal bovine serum (HyClone, Logan, UT, FBS SH30910.03 lot #AXJ47554)) and LIF. mES cells were fed by replacing half the medium with fresh medium daily and passaged every 2 days by trypsinization. Human U2OS osteosarcoma cells (Research Resource Identifier: RRID:CVCL_0042; a gift from David Spector's lab, Cold Spring Harbor Laboratory) were grown in low-glucose DMEM with 10% FBS (full recipe: 500 mL DMEM (ThermoFisher #10567014), 50 mL fetal bovine serum (HyClone FBS SH30910.03 lot #AXJ47554) and 5 mL Penicillin-streptomycin (ThermoFisher #15140122)) and were passaged every 2–4 days before reaching confluency. For live-cell imaging, the medium was identical except DMEM without phenol red was used (ThermoFisher #31053028). Both mouse ES and human U2OS cells were grown in a Sanyo copper alloy IncuSafe humidified incubator (MCO-18AIC(UV)) at 37°C/5.5% $CO_2$.

For all single-molecule experiments (both live and fixed), cells we grown overnight on 25 mm circular no 1.5H cover glasses (Marienfeld, Germany, High-Precision 0117650). Prior to all experiments, the cover glasses were plasma-cleaned and then stored in isopropanol until use. For U2OS cell lines, cells were grown directly on the cover glasses and for mouse ES cells, the cover glasses were coated with Corning Matrigel matrix (Corning #354277; purchased from ThermoFisher #08-774-552) according to manufacturer's instructions just prior to cell plating. After overnight growth, cells were labeled with the relevant Halo- or SNAP-dye at the indicated concentration for 15 min (Halo) or 30 min (SNAP) and washed twice (one wash: medium removed; PBS wash; replenished with fresh medium). At the end of the final wash, the medium was changed to phenol red-free medium keeping all other aspects of the medium the same.

For FRAP experiment, cell preparation was identical except cells where grown on glass-bottom (thickness #1.5) 35 mm dishes (MatTek, Ashland, MA, P35G-1.5–14 C), either directly (U2OS) or Matrigel coated (mESC).

Mouse ES cell lines stably expressing H2B-Halo, H2B-SNAPf, Fucci reporters or Halo-3xNLS were generated using PiggyBac transposition and drug selection. Briefly, the relevant gene (e.g. H2B-Halo) was cloned into a PiggyBac vector co-expressing a drug resistance gene (G418 or Puromycin) and this vector was then co-transfected together with a SuperPiggyBac transposase vector into the relevant mouse ES cell line using Lipofectamine 3000 according to manufacturer's instructions (2 µg expression vector and 1 µg PiggyBac transposase vector per well in a 6-well plate). The following day, selection was then started by adding 1 mg/mL G418 or 5 µg/mL puromycin. An untransfected cell line was selected in parallel and selection was judged to be complete once no live cells were left in the untransfected cell line. For human U2OS cells, stable cell lines were generated by random integration by transfecting the relevant expression vector with drug selection without using the PiggyBac system. Selection was performed in the same way as for mouse ES cells.

### CRISPR/Cas9-mediated genome editing

Knock-in cell lines were created roughly according to published procedures (*Ran et al., 2013*), but exploiting the HaloTag and $SNAP_f$-Tag to FACS for edited cells. The $SNAP_f$-Tag is an optimized version of the SNAP-Tag, and we purchased a plasmid encoding this gene from NEB (NEB, Ipswich, MA, #N9183S). We transfected both U2OS and mES cells using Lipofectamine 3000 (ThermoFisher L3000015) according to manufacturer's protocol, co-transfecting a Cas9 and a repair plasmid (2 µg repair vector and 1 µg Cas9 vector per well in a 6-well plate; 1:2 w/w). The Cas9 plasmid was slightly modified from that distributed from the Zhang lab (*Ran et al., 2013*): 3xFLAG-SV40NLS-pSpCas9 was expressed from a CBh promoter; the sgRNA was expressed from a U6 promoter; and mVenus was expressed from a PGK promoter. For the repair vector, we modified a pUC57 plasmid to contain the tag of interest (e.g. Halo or $SNAP_f$) flanked by ~500 bp of genomic

homology sequence on either side. For N-terminal FLAG-Halo-tagging of mouse *Ctcf* and human *CTCF*, we introduced synonymous mutations (mCTCF: first nine codons after ATG; hCTCF: first 12 codons after ATG), where possible, to prevent the Cas9-sgRNA complex from cutting the repair vector. For C-terminal tagging of mouse *Rad21* with SNAP$_f$-V5, this was not possible. Instead, we designed sgRNAs that overlapped with the STOP codon and, thus, that would not cut the repair vector. For Halo-hCTCF and Halo-mCTCF, we used a TEV linker sequence (EDLYFQS) to link the Halo protein to CTCF; for mRad21, we used the Sheff and Thorn linker (GDGAGLIN) (*Sheff and Thorn, 2004*).

In each case, we designed three or four sgRNAs using the Zhang lab CRISPR design tool (http://tools.genome-engineering.org), cloned them into the Cas9 plasmid and co-transfected each sgRNA-plasmid with the repair vector individually. 18–24 hr later, we then pooled cells transfected with each of the sgRNAs individually and FACS-sorted for YFP (mVenus) positive, successfully transfected cells. YFP-sorted cells were then grown for 4–12 days, labeled with 500 nM Halo-TMR (Halo-Tag knock-ins) or 500 nM SNAP-JF646 (SNAP$_f$-Tag knock-in) and the cell population with significantly higher fluorescence than similarly labeled wild-type cells, FACS-selected and plated at very low density (~0.1 cells per mm$^2$; mES cells) or sorted individually into 96-well plates (U2OS cells). Clones were then expanded and genotyped by PCR using a three-primer PCR (genomic primers external to the homology sequence and an internal Halo or SNAP$_f$ primer). Successfully edited clones were further verified by PCR with multiple primer combinations, Sanger sequencing and Western blotting. We isolated ~6–10 homozygous knock-in clones for each line. The clones chosen for further study all showed similar tagged protein levels to the endogenous untagged protein in wild-type controls.

Sequences for primers and sgRNAs are given in *Supplementary file 2*. All plasmids used in this study, including for genome-editing and transient transfections, are available upon request.

## Teratoma assays

To verify that genome-edited mES cell lines remain pluripotent, we performed teratoma assays and compared wild-type and C59 FLAG-Halo-mCTCF; mRad21-SNAP$_f$-V5 knock-in cells. Briefly, 350,000 cells were injected into the kidney capsule and testis of two 8-week-old Fox Chase SCID-beige male mice (Charles River). Tumors were harvested 27 or 33 days post-injection, fixed with 10% formalin overnight, embedded in paraffin and cut into 5 μm sections and haematoxylin and eosin staining performed. Teratoma assays were performed by Applied Stem Cell, Inc (Milpitas, CA).

## Pathogen testing and cell line authentication

Wild-type and double FLAG-Halo-mCTCF / mRad21-SNAP$_f$-V5 knock-in mouse ES cell line clone 59 were pathogen tested using the IMPACT II test, which was performed by IDEXX BioResearch (Westbrook, ME). Both the wild-type and C59 cell line were negative for all pathogens including Ectromelia, EDIM, LCMV, LDEV, MAV1, MAV2, mCMV, MHV, MNV, MPV, MVM, *Mycoplasma pulmonis*, *Mycoplasma sp.*, Polyoma, PVM, REO3, Sendai, and TMEV. U2OS cell lines were pathogen tested for mycoplasma using a PCR-based assay as described (*Young et al., 2010*) (wild-type U2OS) and pathogen tested for mycoplasma using an imaging assay (DAPI staining; C32 knock-in cell line). Both were negative for mycoplasma. Both mouse ES cells and human U2OS cells were authenticated by whole-genome sequencing and morphology (U2OS morphology was compared to U2OS cells obtained from ATCC). The wild-type and C32 FLAG-Halo-hCTCF knock-in cell lines were further authenticated using Short Tandem Repeat (STR) profiling (performed by Dr. Alison N. Killilea at the UC Berkeley Cell Culture Facility) against the following loci: THO1, D5S818, D13S317, D7S820, D16S539, CSF1PO, AMEL, vWA and TPOX. Both the wild-type and C32 U2OS cell lines showed a 100% match with U2OS.

## Single-molecule imaging

All single-molecule imaging experiments (live-cell residence time measurements, live-cell paSMT at 225 Hz, fixed-cell PALM and fixed-cell dSTORM) were conducted on a custom-built Nikon (Nikon Instruments Inc., Melville, NY) TI microscope equipped with a 100x/NA 1.49 oil-immersion TIRF objective (Nikon apochromat CFI Apo TIRF 100x Oil), EM-CCD camera (Andor, Concord, MA, iXon Ultra 897), a perfect focusing system to correct for axial drift and motorized laser illumination (Ti-TIRF, Nikon), which allows an incident angle adjustment to achieve highly inclined and laminated

optical sheet illumination (*Tokunaga et al., 2008*). The incubation chamber maintained a humidified 37°C atmosphere with 5% $CO_2$ and the objective was similarly heated to 37°C for live-cell experiments. Excitation was achieved using the following laser lines: 561 nm (1 W, Genesis Coherent, Santa Clara, CA) for JF549/PA-JF549 and TMR dyes; 633 nm (1 W, Genesis Coherent) for JF646/PA-JF646 dyes; 405 nm (140 mW, OBIS, Coherent) for all photo-activation experiments. The excitation lasers were modulated by an acousto-optic Tunable Filter (AA Opto-Electronic, France, AOTFnC-VIS-TN) and triggered with the camera TTL exposure output signal. The laser light is coupled into the microscope by an optical fiber and then reflected using a multi-band dichroic (405 nm/488 nm/ 561 nm/633 nm quad-band, Semrock, Rochester, NY) and then focused in the back focal plane of the objective. Fluorescence emission light was filtered using a single band-pass filter placed in front of the camera using the following filters: TMR and JF549/PA-JF549: Semrock 593/40 nm band-pass filter; JF646/PA-JF646: Semrock 676/37 nm bandpass filter. The microscope, cameras, and hardware were controlled through the NIS-Elements software (Nikon).

For simultaneous two-color experiments (dSTORM and PALM experiments), a custom-built setup using two cameras (both Andor iXon Ultra 897 EM-CCD) was used. Cameras were synchronized using a National Instruments (Austin, TX) DAQ board (NI-DAQ PCI-6723). A single-edge dichroic beamsplitter (Di02-R635−25 × 36, Semrock) was used to separate two ranges of wavelengths of emission fluorescence. A 676/37 nm band-pass filter (FF01-676/37-25, Semrock) was placed in front of the first camera and 593/40 nm bandpass filter (FF01-593/40-25, Semrock) in front of the second camera.

In 'slow-tracking' experiments, to measure residence times, long exposure times (300 ms, 500 ms or 800 ms) and low constant illumination laser intensities (to minimize photobleaching) were used. The camera settings were as follows: normal mode; vertical shift speed: 3.3 μs; ROI: variable. Generally, each experiment lasted 20 min per cell corresponding to 4000 frames with a 300 ms exposure time, 2400 frames with a 500 ms exposure time and 1500 frames with an exposure time of 800 ms. We recorded 20 min movies from ~6 cells per cell line or condition per day as well as 6 H2B-Halo cells for the photobleaching correction on the same day and all data presented are from at least three independent experiments conducted on different days.

In 'fast-tracking' stroboscopic paSMT experiments at ~225 Hz, both the main excitation laser (633 nm for PA-JF646 or 561 nm for PA-JF549) and the photo-activation laser (405 nm) were pulsed. Each frame consisted of a 4-ms camera exposure time followed by a ~447 μs camera 'dead' time. The main excitation laser (633 nm) was pulsed for 1 ms starting at the beginning for the 4 ms camera exposure time. The photo-activation laser (405 nm) was pulsed during the ~447 μs camera 'dead' time, to minimize fluorescent background signal. This sequence was verified using an oscilloscope. The camera settings were as follows: frame transfer mode; vertical shift speed: 0.9 μs; ROI: height 90 pixels, width variable. Each cell was imaged for 20,000 frames corresponding to ~1.5 min. The photo-activation laser power was optimized to keep an average molecule density of ~0.5 localizations per frame, corresponding to ~10,000 localization per cell per movie on average. Maintaining a very low density of molecules is necessary to avoid tracking errors. The main excitation laser was used at maximal power. We recorded movies for eight cells per cell line or condition per day, and all data presented are from at least three independent experiments conducted corresponding to at least 24 cells and at least 100,000 localizations.

In PALM experiments, continuous illumination was used for both the main excitation laser (633 nm for PA-JF646 or 561 nm for PA-JF549) and the photo-activation laser (405 nm). However, the intensity of the 405 nm laser was gradually increased over the course of the illumination sequence to image all molecules and at the same time avoid too many molecules being activated at any given frame. The following camera settings were used: 25 ms exposure time; frame transfer mode; vertical shift speed: 0.9 μs; ROI: variable. In total, 40,000–60,000 frames were recorded for each cell (~20–25 min), which was sufficient to image and bleach all labeled molecules. After overnight growth on 25 mm plasma-cleaned coverslips and dye labeling and washings, cells were fixed in 4% PFA in PBS for 20 min at 37°C, washed with PBS and then imaged in PBS with 0.01% (w/v) $NaN_3$ on the same day. All PALM images were acquired at room temperature. All analyses presented contain data from at least 20 cells imaged in at least three independent experiments conducted on different days.

For two-color dSTORM experiments, cell preparation was similar to PALM. After overnight growth on 25 mm plasma-cleaned coverslips and dye labeling and washings, cells were fixed in 4% PFA in PBS for 20 min at 37°C and washed with PBS. We then added 100 nm fluorescent Tetraspeck

beads (diluted 1:1000 in PBS; T7279 ThermoFisher Scientific), allowed the beads to settle and washed three times with PBS. The coverslips were then stored in PBS with 0.01% (w/v) NaN$_3$ until imaged later on the same day. C59 Halo-mCTCF / mRad21-SNAP$_f$ mouse ES cells were labeled with 500 nM Halo-JF646 and 500 nM cp-JF549. mES cells stably expressing H2B-SNAP$_f$ were transfected with a plasmid encoding Halo (only; without being fused to anything) and a GFP-NLS protein used for nuclear demarcation. These cells were similarly labeled. Just before imaging, a STORM imaging buffer (very similar to [*Boettiger et al., 2016*]) was made by mixing 400 μL 50 mM NaCl, 200 mM Tris pH 7.9 with 150 μL 50% glucose solution (w/v), 15 μL GLOX solution, 7.5 μL COT solution and 50 μL MEA solution. The GLOX solution was made by mixing 100 μL 50 mM NaCl, 200 mM Tris pH 7.9 with 7 mg Glucose Oxidase (Sigma-Aldrich) and 25 μL catalase (16 mg/mL). This solution was made the day before imaging. COT solution was made by dissolving 20.8 mg of Cyclooctatetraene (Sigma-Aldrich 138924–1g) in 1 mL DMSO. COT solution aliquots were stored at −20°C and a fresh aliquot used each time. MEA solution was made by dissolving 77 mg cysteamine (Sigma-Aldrich) in 1 mL water. A few drops of 1 M HCl were added to dissolve the cysteamine. STORM imaging buffer was added to the coverslip with fixed cells, the imaging chamber sealed with parafilm and then immediately loaded on the microscope. Both JF549 and JF646 could be converted into a rapidly blinking state in STORM buffer upon high-intensity laser illumination. For each cell, we exposed cells to high-power 405 nm, 561 nm and 633 nm excitation for ~5–10 s. We then acquired 50,000 frames of simultaneous two-color images with constant low-intensity 405 nm excitation and high-intensity 561 nm and 633 nm excitation using 25 ms exposure time on both EM-CCD cameras (Andor iXon Ultra 897). Before imaging, we aligned the two cameras using fluorescent beads (100 nm TetraSpeck beads; T7279 ThermoFisher Scientific) to a registration offset below 50 nm. Before imaging each cell, we imaged a cell-adjacent bead. Similarly, after imaging each cell we also imaged a different cell-adjacent bead (1000 frames at 25 ms each time). We then used the mean offset from the bead measurements before and after imaging a cell for two-color registration for that cell. We estimate a chromatic shift registration error of ~10 nm. The pair cross correlation data presented are from around ~12–18 cells measured on 3 different days. All PALM and dSTORM experiments on fixed cells were conducted at room temperature to minimize drift.

## Analysis of single-molecule images

All single-molecule imaging data were processed using a custom-written MATLAB implementation of the MTT algorithm (*Sergé et al., 2008*). A GUI of this implementation, SLIMfast (*Normanno et al., 2015*), is available at https://elifesciences.org/content/5/e22280/supp-material1 (*Teves et al., 2016*). Briefly, single molecules are localized using bi-dimensional Gaussian fitting (approximating the microscope PSF) subject to a generalized log-likelihood ratio test with a 'localization error' threshold (in the range of $10^{-6}$-$10^{-7}$), with the option of allowing deflation to detect molecules partially obscured by others. Tracking, that is connecting localizations between consecutive frames, was limited by setting a maximal expected diffusion constant, and takes the trajectory history into account as well as allowing for gaps due to blinking or missed localizations.

For analysis of 'slow-tracking' experiments, to measure residence times, the following algorithm parameters were used: Localization error: $10^{-7}$; deflation loops: 1; Blinking (frames): 2; maximum number of competitors: 1; maximal expected diffusion constant (μm$^2$/s): 0.1.

For analysis of 'fast-tracking' stroboscopic paSMT experiments at ~225 Hz, the following algorithm parameters were used: Localization error: $10^{-6.25}$; deflation loops: 0; Blinking (frames): 1; maximum number of competitors: 3; maximal expected diffusion constant (μm$^2$/s): 20.

For analysis of PALM experiments, the following algorithm parameters were used: Localization error: $10^{-6}$; deflation loops: 0; Blinking (frames): 1; maximum number of competitors: 3; maximal expected diffusion constant (μm$^2$/s): 0.05.

For analysis of dSTORM experiments, we used the same algorithm parameters as for PALM analysis for both color channels.

All subsequent analyses of trajectories were performed using custom-written code in MATLAB as described in detail in the following sections.

## Kinetic modeling of fast 225 Hz SMT data

To extract kinetic information from fast stroboscopic paSMT at approximately 225 Hz, we developed and fit a mathematical model to the jump length or displacement distributions. Our approach is largely inspired by an elegant modeling approach previously introduced by Mazza *et al.* (*Mazza et al., 2012*), but with a number of significant differences and modifications that we will highlight below.

The evolution over time of a concentration of particles located at the origin as a Dirac delta function and which follows free diffusion in two dimensions with a diffusion constant $D$ can be described by a propagator (also known as Green's function). Properly normalized, the probability of a particle starting at the origin ending up at a location $r = (x, y)$ after a time delay, $\Delta\tau$, is then given by:

$$P(r, \Delta\tau) = N \frac{r}{2D\Delta\tau} e^{-\frac{r^2}{4D\Delta\tau}}$$

Here, $N$ is a normalization constant with units of length. In practice, we compare this distribution to binned data. Thus, in practice, we integrate this distribution over a small histogram bin window, $\Delta r$, to obtain a normalized distribution to compare to the empirically measured distribution. For simplicity, we therefore leave out this normalization constant of subsequent expressions.

Furthermore, in practice, we are unable to determine the precise localization of a single molecule. Instead, it is associated with a certain localization error, $\sigma$, which under our stroboscopic paSMT conditions is approximately 35 nm. Correcting for localization errors is important because it will otherwise appear as if molecules move further between frames than they actually did. Thus, we obtain the following expression for the jump length distribution taking localization error, $\sigma$, into account (*Matsuoka et al., 2009*):

$$P(r, \Delta\tau) = \frac{r}{2(D\Delta\tau + \sigma^2)} e^{-\frac{r^2}{4(D\Delta\tau + \sigma^2)}}$$

DNA-binding molecules such as CTCF can generally exist in either a bound or a freely diffusing state. The bound state exhibits very short jump lengths (presumably due to slow chromatin diffusion) and has an associated diffusion constant, $D_{\mathrm{BOUND}}$, whereas the freely diffusing population tends to exhibit much longer jump lengths and has its own associated diffusion constant, $D_{\mathrm{FREE}}$. Next, we assume that binding to chromatin and unbinding from chromatin are both first-order processes with rate constants $k_{\mathrm{ON}}^*$ and $k_{\mathrm{OFF}}$. We denote $k_{\mathrm{ON}}^*$ with a '*' because it is really a pseudo first-order process since it depends on the concentration of free binding sites: $k_{\mathrm{ON}}^* = [BS_{\mathrm{FREE}}]k_{\mathrm{ON}}$. Thus, the steady-state jump length distribution of a population of molecules that can exist in either their bound or free state is then given by:

$$P(r, \Delta\tau) = F_{\mathrm{BOUND}} \frac{r}{2(D_{\mathrm{BOUND}}\Delta\tau + \sigma^2)} e^{-\frac{r^2}{4(D_{\mathrm{BOUND}}\Delta\tau + \sigma^2)}} + (1 - F_{\mathrm{BOUND}}) \frac{r}{2(D_{\mathrm{FREE}}\Delta\tau + \sigma^2)} e^{-\frac{r^2}{4(D_{\mathrm{FREE}}\Delta\tau + \sigma^2)}}$$

where $F_{\mathrm{BOUND}}$ is the fraction of the population that is bound to chromatin and, $F_{\mathrm{FREE}} = 1 - F_{\mathrm{BOUND}}$, is the fraction of the population that is exhibiting free 3D diffusion. These fractions are related to the first-order rate constants:

$$F_{\mathrm{BOUND}} = \frac{k_{\mathrm{ON}}^*}{k_{\mathrm{ON}}^* + k_{\mathrm{OFF}}}$$

$$F_{\mathrm{FREE}} = (1 - F_{\mathrm{BOUND}}) = \frac{k_{\mathrm{OFF}}}{k_{\mathrm{ON}}^* + k_{\mathrm{OFF}}}$$

These expressions assume that molecules do not change between their bound and free states during the time delay between frames, $\Delta\tau$. Previous studies have derived analytical expressions to account for this (*Mazza et al., 2012*; *Yeung et al., 2007*). However, implementing these expressions numerically greatly slows down fitting the model to the raw jump length distributions. Accounting for state-changes between the free and bound states was necessary in the previous study by Mazza *et al.* (*2012*) because relatively long exposure times (40 ms or 25 Hz) and lag times, $\Delta\tau$, (up to 800 ms) were considered. In this study, we are imaging at a much higher frame-rate (4.4477 ms exposure

or ~225 Hz) and only consider much shorter lag times, $\Delta\tau$, (up to seven jumps, i.e. 31.5 ms). Thus, in our case, the probability of observing a state-change is much lower. Moreover, the residence time of CTCF (~60–75 s) is much longer than the residence time of p53 (~1.8 s) (*Mazza et al., 2012*). Thus, we can calculate the probability that a bound CTCF molecule unbinds during the longest lag times considered ($\Delta\tau$ = 31.5 ms) as:

$$P_{\text{SWITCH}} = 1 - e^{-k_{\text{OFF}}\Delta\tau} \approx 7 \cdot 10^{-5}$$

Thus, accounting for state changes during the lag time, $\Delta\tau$, makes a negligible difference for CTCF. Even if we consider short-lived non-specific interactions, the probability of a state-change is still negligible with our short lag times.

Single-molecule tracking (SMT) is heavily biased toward bound molecules and against freely diffusing molecules for two major reasons. First, almost all single-molecule localization algorithms, including the MTT-algorithm (*Sergé et al., 2008*) used here, achieve sub-diffraction limit resolution (super-resolution) by treating individual fluorophores as point-source emitters, which generate blurred images that are described by the Point-Spread Function (PSF) of the microscope. Two-dimensional Gaussian modeling of the PSF allows extraction of the particle centroid with sub-pixel resolution. In SMT experiments, this works well for bound molecules, which exhibit negligible movement during the laser exposure time. However, fast moving molecules will tend to 'motion-blur' because they can move several pixels during the long exposure times typically used in SMT experiments. 'Motion-blurred' particles will thus spread their photons over multiple pixels in the direction of their movement. Therefore, they tend to be missed by most PSF-fitting localization algorithms, which results in a large bias toward bound molecules and a general bias against fast-moving molecules. This means that the bound fraction will be overestimated. To minimize this bias against fast-moving molecules, we use stroboscopic illumination where although we have a time delay of $\Delta\tau$ = 4.4477 ms, we only laser-illuminate the sample for 1 ms per frame. For a molecule like CTCF where the freely diffusing population has an apparent $D_{\text{FREE}}$ ~2.5 $\mu m^2$/s, we can calculate the fraction of the population which moves more than a certain length during the 1 ms laser illumination time. Using our imaging setup (pixel size: 160 nm), less than ~0.0036% (~3.6 molecules per 100,000 molecules) of the free CTCF population move more than two pixels during the 1 ms laser exposure time. Thus, while we cannot eliminate all bias against moving molecules, our fast stroboscopic SMT methods greatly reduce bias against fast-moving molecules compared to previous approaches.

Second, fast-moving molecules are likely to move out of the focal plane or axial detection window ($\Delta z$) during 2D image acquisition. Even though we consider short lag times $\Delta\tau$ ~4.5–31.5 ms, this is still long enough for a large fraction of the free population to be lost. As a consequence, bound molecules tend to have much longer trajectories than do free molecules. Again, this means that we are oversampling the bound population and undersampling the free population. To correct for this, we consider the probability that a freely diffusing molecule with diffusion constant, $D_{\text{FREE}}$, will move out of the axial detection window, $\Delta z$, during a lag time, $\Delta\tau$. This problem has also been previously considered by Kues and Kubitscheck (*Kues and Kubitscheck, 2002*). If we consider the extreme case of a population of molecules equally distributed one-dimensionally along an axis, $z$, with an absorbing boundary at $Z_{\text{MAX}} = \Delta Z/2$ and $Z_{\text{MIN}} = -\Delta Z/2$, the fraction of molecules remaining at lag time, $\Delta\tau$, is given by:

$$P_{\text{LEFT}}(\Delta\tau) = \frac{1}{\Delta z} \int_{-\Delta z/2}^{\Delta z/2} \left\{ 1 - \sum_{n=0}^{\infty} (-1)^n \left[ \text{erfc}\left( \frac{\frac{(2n+1)\Delta z}{2} - z}{\sqrt{4D_{\text{FREE}}\Delta\tau}} \right) + \text{erfc}\left( \frac{\frac{(2n+1)\Delta z}{2} + z}{\sqrt{4D_{\text{FREE}}\Delta\tau}} \right) \right] \right\} dz$$

However, this expression significantly overestimates how many freely diffusing molecules are lost since it assumes absorbing boundaries – any molecules that comes into contact with the boundary at $\pm \Delta z/2$ are permanently lost. In reality, there is a significant probability that a molecule, which has briefly contacted or exceeded the boundary, re-enters the axial detection window, $\Delta z$, during a lag time, $\Delta\tau$. Moreover, since we allow trajectory gaps of one during in our tracking algorithm (i.e. a molecule present in frame $n$ and $n+2$ can still be tracked even if it was not localized in frame $n+1$), we must consider the probability that a lost molecule re-enters the axial detection window during twice the lag time, $2\Delta\tau$. This results in the somewhat counter-intuitive effect, which was also noted by Kues and Kubitscheck, that the decay rate depends on the microscope frame rate – in other words, the

fraction lost depends on how often one 'looks'. One approach (*Mazza et al., 2012*) of accounting for this is to use a corrected axial detection window larger than the true axial detection window: $\Delta z_{\mathrm{CORR}} > \Delta z$.

To find the corrected axial detection window, we first measured the true empirical axial detection window, $\Delta z$. We labeled C59 Halo-mCTCF mouse embryonic stem cells and C32 Halo-hCTCF human U2OS cells grown on plasma-cleaned 25 mm #1.5 cover glasses with JF646 at a low enough density to clearly observe single molecules and fixed them in 4% PFA in PBS for 20 min. We then collected an extensive z-stack throughout the nucleus with a range of 6 µm and a step size of 20 nm (301 frames) and imaged single molecules at a signal-to-background ratio comparable to the one used during our fast 225 Hz paSMT experiments. We tracked molecules using the MTT algorithm (*Sergé et al., 2008*) and the same parameters used for our paSMT experiments. We then analyzed the survival curve, corrected for photobleaching, of single JF646-labeled Halo-CTCF molecules as a function of the step size and found the axial detection window to be approximately $\Delta z \approx 700$ nm and highly similar in U2OS and mES cells under HiLo-illumination (*Tokunaga et al., 2008*).

Next, we performed Monte Carlo simulations following the Euler-Maruyama scheme. For a given diffusion constant, $D$, we randomly distributed 50,000 molecules one-dimensionally along the z-axis from $Z_{\mathrm{MIN}} = -\Delta z/2 = -350$ nm to $Z_{\mathrm{MAX}} = \Delta z/2 = 350$ nm, where $\Delta z \approx 700$ nm. Next, using a time-step of $\Delta \tau = 4.4477$ ms, we simulated one-dimensional Brownian diffusion along the z-axis by randomly picking Gaussian-distributed numbers from a normal distribution with parameters: $\mu = 0$; $\sigma = \sqrt{2D\Delta\tau}$ using the function normrnd in MATLAB. For time gaps from 1 $\Delta\tau$ to 15 $\Delta\tau$, we then calculated the fraction of molecules that were lost, allowing for one missing frame as in our tracking algorithm. We repeated these simulations for particles with diffusion constants in the range of $D = 1$ µm²/s to $D = 12$ µm²/s to generate a comprehensive dataset over a range of biologically plausible diffusion constants. We then performed least-squares fitting of this dataset to the equation for $P_{\mathrm{LEFT}}(\Delta\tau)$ using a corrected $\Delta z_{\mathrm{CORR}}$:

$$\Delta z_{CORR} = \Delta z + a\sqrt{D} + b$$

The simulated data were well fit using this corrected axial detection window, and we found the following best-first parameters: $a = 0.15716$ s$^{-1/2}$; $b = 0.20811$ µm. Practically, we evaluated the equation for $P_{\mathrm{LEFT}}(\Delta\tau)$ using numerical integration in MATLAB and aborted the infinite sum once the absolute value of another iteration fell below $10^{-12}$. We performed non-linear least-squares fitting in MATLAB by stochastically generating random parameter guesses for $a$ and $b$ as a starting point for the least-squares fitting routine lsqcurvefit and iterating using multiple random input guesses to avoid local minima.

Having derived an analytical expression for the probability of a free molecule being lost due to axial diffusion during the imaging time, we can now thus write down the final equations used for fitting the raw jump length distributions:

$$P(r, \Delta\tau) = F_{\mathrm{BOUND}} \frac{r}{2(D_{\mathrm{BOUND}}\Delta\tau + \sigma^2)} e^{-\frac{r^2}{4(D_{\mathrm{BOUND}}\Delta\tau + \sigma^2)}} + Z_{\mathrm{CORR}}(\Delta\tau)(1 - F_{\mathrm{BOUND}}) \frac{r}{2(D_{\mathrm{FREE}}\Delta\tau + \sigma^2)} e^{-\frac{r^2}{4(D_{\mathrm{FREE}}\Delta\tau + \sigma^2)}}$$

where:

$$Z_{\mathrm{CORR}}(\Delta\tau) = \frac{1}{\Delta z} \int_{-\Delta z/2}^{\Delta z/2} \left\{ 1 - \sum_{n=0}^{\infty} (-1)^n \left[ \mathrm{erfc}\left( \frac{\frac{(2n+1)\Delta z}{2} - z}{\sqrt{4D_{\mathrm{FREE}}\Delta\tau}} \right) + \mathrm{erfc}\left( \frac{\frac{(2n+1)\Delta z}{2} + z}{\sqrt{4D_{\mathrm{FREE}}\Delta\tau}} \right) \right] \right\} dz$$

and:

$$\Delta z = 0.700\ \mu m + 0.15716\ s^{-1/2}\sqrt{D} + 0.20811\ \mu m$$

In practical terms, we consider the jump length or displacement distributions for timepoints 1 to 8, corresponding to seven jumps with delays from 1 $\Delta\tau$ to 7 $\Delta\tau$ (i.e. this includes 6 jumps of 1 $\Delta\tau$, 5 jumps of 2 $\Delta\tau$, and so on). Thus, the probability of seeing a free molecule present in the first frame is higher in the second frame than in the seventh frame according the $Z_{\mathrm{CORR}}$ equation above. While we have many trajectories that are much longer than eight localizations, we refrain from using the entire trajectories since almost all very long trajectories (e.g. >100 localizations) are highly biased toward

bound molecules. While the above $Z_{\mathrm{CORR}}$ equation should in principle correct for this, at long time lags the probability of still seeing a moving molecule approaches zero and thus small errors in the $Z_{\mathrm{CORR}}$ equation, which is an approximation, is likely to strongly affect the estimation of the bound fraction.

We note that a question arises of whether to use the entire trajectory or not. One bias against moving molecules is that frequently, freely diffusing molecules will translocate through the axial detection window, $\Delta z$, yielding only a single detectable localization and thus no jumps to be counted. Conversely, one bias against bound molecules, is that moving molecules can re-enter the axial detection window multiple times resulting in the same molecule appearing as multiple distinct trajectories and thus being over-counted. Clearly, the extent of the bias will depend on the photobleaching rate – in the limit of no photobleaching, a single freely diffusing molecule could yield a very high number of different trajectories, leading to large over-counting of the free population. However, in practice, under our stroboscopic paSMT conditions, the average dye lifetime is quite short. We note that dye disappearance is both due to photobleaching and blinking, but note that blinking should not affect estimates of the fraction bound. The actual mean number of frames depends on the fraction bound and diffusion constant – proteins with slow diffusion constants and a high bound fraction stay in the axial detection volume for longer and thus yield longer trajectories. Accordingly, for Halo-mCTCF, the mean number of frames per trajectory is ~3–4, whereas for Halo-3xNLS it is less than two, even though the photobleaching rate is the same. We took two approaches to test whether the fraction of the trajectory that is included in the modeling would strongly affect the fraction bound estimate: analysis of our raw data and Monte Carlo simulations according to the Euler-Maruyama scheme. First, in the case of our raw data, the difference between using only the first seven jumps and using the entire trajectory only affects the fraction bound estimate by a few percentage points, suggesting that it makes a minor difference under conditions where photobleaching and blinking results in relatively short trajectories. Second, we performed Monte Carlo simulations following the Euler-Maruyama scheme and with the following assumptions: 50% of molecules are bound and the free diffusion constant is 2.5 μm²/s; the axial detection volume is 700 nm and the laser excitation beam under highly inclined and laminated optical sheet illumination (HiLo) illuminates ~4 μm (*Tokunaga et al., 2008*), corresponding to half the nucleus (nuclear diameter: 8 μm); molecules within the HiLo sheet photobleach with a constant rate (thus molecules can photobleach outside of the detection slice as in our experiments); the 2D localization error is 35 nm and the timestep is 4.5 ms; since the vast majority of trajectories lasts no more than tens of milliseconds, but both the CTCF unbinding rate (~1 min) and re-binding rate (~1 min) are much slower, we ignore changes in state (bound vs. free) during the trajectory lifetime; Brownian motion was simulated for 500,000 trajectories in three dimensions enclosed within the nucleus by picking random numbers in each dimension from a normal distribution defined as: $N \sim \left(0, \sqrt{2D\Delta\tau}\right)$. Our simulations showed that our paSMT modeling approach could accurately infer both the free diffusion constant (slight overestimate of $D$, but error less than 5%) and the fraction bound and that using the entire trajectory leads to a very small overestimate of the bound fraction (one percentage point) and that using the first seven jumps only leads a small underestimate of the bound fraction (~3 percentage points) under conditions where the mean trajectory length (~3) was similar to the mean trajectory length for Halo-mCTCF in mESCs under our experimental conditions. However, under conditions with negligible photobleaching and extremely long trajectories of a mean length of ~100 frames, using only the first seven jumps leads to a serious underestimate of the bound fraction. We note that it is not experimentally realistic to obtain trajectories of this length with currently available dyes and microscope modalities and thus not relevant in this case, but we nevertheless note that generalizing the approach to trajectories of any length is an interesting future direction. Finally, because of the numerous other biases against free molecules noted above, we only use the first seven jumps and ignore all subsequent jumps in longer trajectories for our model fitting in this case.

We then fit the above equation for, $P(r, \Delta\tau)$, to the raw jump lengths distributions for time gaps of $1\Delta\tau$ to $7\Delta\tau$ corresponding to 4.5 ms to 31.5 ms. Although we show the fit function to the probability density, that is histograms (*Figure 3A–E*), since this is more intuitive, this introduces binning artifacts (bin: 10 nm). Thus, for quantitative analysis, we instead fit the model to the cumulative distribution function (CDF) calculated from the data. The model has three fit parameters, $D_{\mathrm{BOUND}}$, $D_{\mathrm{FREE}}$ and $F_{\mathrm{BOUND}}$, and is fit to the combined jump length CDFs (from $1\Delta\tau$ to $7\Delta\tau$) using least squares

fitting. We constrain $D_{\text{BOUND}}$ to a range of [0.0005, 0.08] µm²/s, but note that slight errors in the estimation of the localization error would make it appear as if the bound molecules move faster or slower than they actually do. $F_{\text{BOUND}}$ is of course constrained to a range of [0, 1] and we only constrain $D_{\text{FREE}}$ to be greater than 0.15 µm²/s. We randomly generated initial parameter guesses for $D_{\text{BOUND}}$, $D_{\text{FREE}}$ and $F_{\text{BOUND}}$ and then fit the model to the seven CDFs through non-linear least squares minimization implemented in MATLAB through the function lsqcurvefit. We then repeat this for multiple iterations of random initial parameter guesses and record the best-fit parameters. Thus, from the kinetic modeling, we obtain $D_{\text{BOUND}}$, $D_{\text{FREE}}$ and $F_{\text{BOUND}}$, from which we can also calculate $F_{\text{FREE}} = 1 - F_{\text{BOUND}}$. We note that although the previous study on p53 by Mazza *et al.* (*2012*) required two freely diffusive states and one bound state to fit the jump length distributions, in our case a single free diffusion state and one bound state were sufficient to accurately fit the raw jump length distributions. Thus, we did not consider the possibility of additional diffusive states.

## Inferring parameters related to the CTCF and Rad21 target search mechanism

Next, we sought to further extend our knowledge of the nuclear target search mechanism in vivo using the parameters inferred from our kinetic modeling of the fast paSMT data as well as our residence time measurements. First, we illustrate the approach using CTCF as an example. We will continue with the steady-state two-state model (bound or free) introduced above, but further distinguish specific and non-specific binding. From the kinetic model fitting above, we determine the total bound fractions for CTCF to be: mESC C59 Halo-mCTCF, 68.0 ± 3.3%; mESC C87 Halo-mCTCF, 68.4 ± 2.1%; U2OS C32 Halo-hCTCF, 58.9 ± 2.0%. However, this total bound fraction contains both CTCF molecules bound specifically to their cognate binding sites and non-specific interactions. For example, sliding on DNA would be indistinguishable from stable binding to a cognate site under our paSMT conditions (localization error ~35 nm). We estimate the fraction that is non-specifically bound using a mutant CTCF, 11ZF-mut-Halo-mCTCF, where we have introduced mutations into the DNA-binding domain. This mutant contains a His-to-Arg mutation in each of the 11 zinc-finger domains. Since the mutant, by design, is unable to interact specifically with chromatin through its zinc-finger domains, we reason that this mutant interacts only non-specifically. From our kinetic model fitting of the 11ZF-mut-Halo-mCTCF jump length histograms, we estimate the bound fraction for this mutant to be 19.1 ± 4.1% in mouse ES cells and 17.7% in human U2OS cells. Thus, the specifically bound fraction can be calculated according to:

$$F_{\text{BOUND, specific}} = F_{\text{BOUND, total}} - F_{\text{BOUND, non-specific}}$$

Using the numbers above, we then obtain the following estimates for the specifically bound fraction: mESC C59 Halo-mCTCF, 48.9%; mESC C87 Halo-mCTCF, 49.3%; U2OS C32 Halo-hCTCF, 41.2%. We note that this estimation is associated with definitional uncertainty as well measurement uncertainty. It is difficult to define exactly what a non-specific interaction is, but it likely involves transient binding and/or sliding on DNA. It is also difficult to define precisely for how long a molecule has to associate with DNA for that to be reasonably counted as a non-specific interaction. Nevertheless, if we operationally define non-specific interaction here as an interaction present after mutation of the DNA-binding domain, we can proceed with investigating the target search mechanism.

Next, we would like to determine the average time it takes a single CTCF protein to find another specific binding site. In the following, we will use 's' and 'ns', as abbreviations for specific and non-specific, respectively. The pseudo-first-order rate constant for specific binding sites, $k_{\text{ON,s}}^*$, is related to the fraction bound by:

$$F_{\text{BOUND,S}} = \frac{k_{\text{ON,s}}^*}{k_{\text{ON,s}}^* + k_{\text{OFF,s}}^*} \Longleftrightarrow k_{\text{ON,s}}^* = \frac{F_{\text{BOUND,s}} k_{\text{OFF,s}}}{1 - F_{\text{BOUND,s}}}$$

We determined the off-rate for a specific interaction in our residence time measurements (*Figure 2*). Thus, from the previously determined values of $F_{\text{BOUND,s}}$ and $k_{\text{OFF,s}}$, we can calculate $k_{\text{ON,s}}^*$. $k_{\text{ON,s}}^*$ is an interesting constant because it is directly related to the average search time for a specific CTCF-binding site:

$$\tau_{\text{search,s}} = \frac{1}{k_{\text{ON,s}}^*} = \frac{1 - F_{\text{BOUND,s}}}{F_{\text{BOUND,s}} k_{\text{OFF,s}}}$$

When we plug in the previously determined values of $F_{BOUND,s}$ and $k_{OFF,s}$, we thus obtain total search times of: mESC C59 Halo-mCTCF,~65.9 s; mESC C87 Halo-mCTCF,~62.6 s; U2OS C32 Halo-hCTCF,~102.8 s. We note that the search times depend sensitively on $k_{OFF,s}$, such that if a CTCF residence time of ~4 min is used instead, the search time also increases to around 4 min in mES cells and to ~5.7 min in U2OS cells. Regardless of the total search time, CTCF molecules spend roughly 50% of their time searching for binding sites in mES cells and roughly 60% of their time searching for binding sites in human U2OS cells. This search time contains intermittent periods of free 3D diffusion interrupted by brief non-specific binding or sliding interactions on chromatin. For example, for mESC C59 Halo-mCTCF, 51.1% of the total time is spent searching - 19.1% of the total time is spent in 1D sliding on DNA or transient interactions and 32.0% of the total time is spent on free 3D diffusion. Since we know the average search time to be ~65.9 s, we can thus calculate that during this average search time, ~41.3 s are spent in free 3D diffusion and ~24.6 s are spent in non-specific DNA interactions such as sliding. Thus, for mESC C59 Halo-mCTCF roughly 37% of the total search time is spent in non-specific DNA interactions and roughly 63% of the time is spent on free 3D diffusion. Similar analysis of C32 Halo-hCTCF in human cells show that 58.8% of the total time is spent searching, with 17.7% of the total time in non-specific chromatin association (e.g. 1D sliding) and 41.1% of the total time in free 3D diffusion. Thus, with an average search time of ~102.8 s, human Halo-hCTCF spends on average ~30.9 s on non-specific chromatin association and ~71.9 s on free 3D diffusion.

We can apply the same approach to cohesin as measured by following mRad21 in mES cells. We note that the above approach assumes a single bound state and a single free state. This is certainly too simplistic in S/G2, since our FRAP experiments suggest that the chromatin residence time of cohesin involved in sister chromatid cohesion is likely much longer than the cohesin involved in chromatin looping. Moreover, it is far from clear that the ON-rate, that is topological loading of cohesin onto chromatin, would be similar for cohesin involved in chromatin looping and in sister chromatid cohesion. Thus, we restrict our analysis to G1. Even then, we stress that this analysis assumes that all topologically engaged G1 cohesin has the same ON- and OFF-rates. We estimated the G1 cohesin residence time to be 19.51 min (C45 mRad21-Halo) and 24.16 min (C59 mRad21-SNAP$_f$). In the following, we will use the mean: 21.8 min. Using stroboscopic paSMT, we estimated the G1 total fraction bound of cohesin to be 53.5 ± 4.1% and the non-specifically bound fraction to be 13.7 ± 3.1% using a mutant (F601R, L605R, Q617K) that is reported to be unable to form cohesin complexes (*Haering et al., 2004*). Thus, 39.8% of cohesin is topologically bound to chromatin, 13.7% non-specifically associated with chromatin and 46.5% in free 3D diffusion in G1-phase of the cell cycle. Non-specific chromatin association may include non-productive topological loading attempts. This yields a search time of ~33.0 min of which around 7.51 min is spent on non-specific chromatin association (e.g. sliding) and 25.49 min is spent on free 3D diffusion. We note that this description of the cohesin search mechanism is somewhat simplified since assisted topological loading is a bit more complicated than finding a cognate-binding site for a typical sequence-specific transcription factor. Rather, it is likely that the cohesin search mechanism is regulated by other protein interaction partners and by post-translational modifications (*Skibbens, 2016*). Nevertheless, even if topological loading involves multiple steps, the process can be described as a single first-order reaction if there is a single rate-limiting step.

## Residence time measurements from SMT

To extract residence times from SMT data recoded at long exposure time, we took a hybrid approach related to that of *Chen et al. (2014)* and *Mazza et al. (2012)*. Briefly, we took advantage of long exposure times (300 ms, 500 ms or 800 ms) as previously described (*Chen et al., 2014*): this causes freely-diffusing molecules to motion-blur into the background such that they are generally missed by our detection algorithm (*Sergé et al., 2008*). We then recorded the trajectory length of each 'bound' molecule and used these to generate a survival curve (1-CDF). However, as previously reported there are multiple contributions to this survival curve beyond specific binding, which is what we are interested in, such as non-specific binding (*Chen et al., 2014*) and slow-diffusing

molecules (*Mazza et al., 2012*). Beyond these two, localization errors can cause both false-positive and false-negative detections. False negative detections especially occur for molecules close to being out-of-focus. This can cause a single long trajectory to appear as many short ones. Thus, we performed double-exponential fitting (corresponding to specific and non-specific binding) using:

$$P(t) = A e^{-k_{\mathrm{ns}} t} + B e^{-k_{\mathrm{s}} t}$$

where $k_{\mathrm{ns}}$ corresponds to the unbinding rate for non-specific binding and $k_{\mathrm{s}}$ corresponds to the unbinding rate constant for specific binding. We note that the first rate constant, $k_{\mathrm{ns}}$, is likely to be contaminated by localization errors (e.g. from molecules close to being out-of-focus) and experimental noise and we therefore caution against over interpreting it. To filter out contributions from tracking errors and slow-diffusing molecules, we applied an objective threshold as previously described to consider only particles tracked for at least $N_{\mathrm{min}}$ frames (*Mazza et al., 2012*). To determine $N_{\mathrm{min}}$, we plotted the inferred residence time as a function of $N_{\mathrm{min}}$ and observed convergence to a single value after ~2.5 s (i.e. 8 frames at 300 ms exposure time, 5 frames at 500 ms exposure time, 3 frames at 800 ms exposure time; *Figure 2—figure supplement 1A*). We thus used this threshold to determine the value of $k_{\mathrm{s}}$. The measured $k_{\mathrm{s}}$, however, reflects both unbinding from chromatin as well as photobleaching etc.:

$$k_{\mathrm{s}} = k_{\mathrm{s,true}} + k_{\mathrm{bias}}$$

Photobleaching clearly needs to be corrected for. But several other factors also contributed faster apparent unbinding. Among these were axial cell drift, lateral cell drift, fluctuating background and others. Axial cell drift can cause a single molecule to move gradually out-of-focus, which appears as unbinding. We also observe significant lateral cell drift, especially for mES cells due to cell movement, which can appear as unbinding if particle movement exceeds the threshold. Drift is especially an issue for molecules exhibiting relatively stable binding such as CTCF, where we occasionally, but very rarely, observe single molecules for around 10 min under constant laser illumination. To correct for all these factors including photobleaching, we reasoned that, if we assume that all these processes are Poisson processes, then the sum of independent Poissons is also a Poisson. If we further assume that these processes will affect H2B-Halo to the same extent as CTCF (i.e. photobleaching depends only on the dye used and the laser intensity; axial chromatin or cell drift is the same for Halo-CTCF cells as for H2B-Halo cells), then we can measure an apparent unbinding rate for H2B-Halo and use this as $k_{\mathrm{bias}}$. This analysis assumes that any apparent unbinding of H2B will be due to photobleaching or drift etc., which is consistent with our FRAP data. However, we note that although H2B molecules are no doubt occasionally evicted from chromatin (e.g. during chromatin remodeling), as long as the rate is much smaller than the unbinding rate of CTCF, this makes a negligible contribution. Thus, to estimate $k_{\mathrm{bias}}$, we repeated the experiments on mES or U2OS cells stably expressing H2B-Halo and estimated $k_{\mathrm{bias}}$ as the slow component from double-exponential fitting as described above. We always performed the H2B-Halo control experiment on the same day as the other experiments. Having measured $k_{\mathrm{bias}}$, we then calculated the residence time as

$$\tau_{\mathrm{s}} = \frac{1}{k_{\mathrm{s,true}}}$$

We note that the above analysis assumes that the unbinding rate for all CTCF sites is identical, which is clearly an approximation, although the ability of the model to fit the data suggests it is a reasonable approximation. However, this analysis would miss a very small CTCF fraction (<3%) showing different residence times. So the above-calculated residence time should be interpreted as an average residence time, which holds for most CTCF sites, but may not hold for all.

## PALM – data processing and clustering analysis

We extracted single-molecule x,y coordinates from single-color PALM images using the following pipeline. We took advantage of the high photostability of the PA-JF549 and PA-JF646 dyes to increase localization accuracy and to perform drift correction. Similar fiducial marker independent drift-correction algorithms have been described previously (*Elmokadem and Yu, 2015*; *Wang et al., 2014*). At the laser intensity used and an exposure time of 25 ms, each JF549/JF646 molecule lasted ~5–10 frames on average before photobleaching. Thus, after localizing molecules in each

frame and tracking them between frames, we obtaining several estimates of the true x,y coordinates for each molecule, which improves the localization precision. Moreover, since each frame contained 5–10 molecules on average this allowed us to perform drift correction by tracking the average drift of particles over time after binning to average out noise in individual localizations.

For spatial clustering analysis, we segmented the nucleus by convolving the PSF with the single-molecule localizations and then blurring the image using iterative Gaussian smoothing followed by thresholding or by manual polygon segmentation. We then divided the nucleus into partially over-lapping 3 μm squares and performed clustering analysis on these squares using a recently reported Bayesian algorithm (*Rubin-Delanchy et al., 2015*). We used the same prior as published (*Rubin-Delanchy et al., 2015*) and performed cluster identification and characterized clusters according to their cluster radius and fraction of molecules in clusters as described (*Rubin-Delanchy et al., 2015*).

A major concern in clustering analysis of PALM images is photo-blinking, where a dye turns off for some frames and the re-appears. Since we track single molecules across frames and allow for gaps of 1 frame, most molecules that exhibit multiple appearances will be collapsed into a single localization. However, it is not possible to unambiguously distinguish two different co-localizing mol-ecules that appear many frames apart, from a single molecule that exhibits a long photo-blink. Thus, to investigate to what extent the apparent clustering that we observe is due to uncorrected photo-blinking we took the following approaches.

First, we compared our Halo-CTCF and Rad21-Halo PALM reconstructions to H2B-Halo and Halo-3xNLS. While there is no known protein whose nuclear organization perfectly exhibits complete spa-tial randomness, we reasoned that Halo-3xNLS should exhibit a relatively uniform distribution. Thus, by using the same dye and imaging conditions as for CTCF and Rad21, we treat the level of cluster-ing observed for Halo-3xNLS as being largely due to blinking, and thus generate a 'blinking floor'. Since both CTCF and Rad21 exhibits much higher clustering than Halo-3xNLS (*Figure 4—figure sup-plement 1C*), we conclude that most of the observed clustering is not due to photo-blinking. We note that both the H2B-Halo and Halo-3xNLS transgenes are expressed at very high levels. Thus, we empirically adjusted the PA-JF549 concentration so as to get similar numbers of localizations as for CTCF, so as to exclude any bias coming from the number of molecules.

Second, in mES cells CTCF and H2B exhibit comparable levels of clustering, but Ripley's $L(r)$-$r$ curves are qualitatively different, with H2B showing clustering at larger length scales. This further suggests that our PALM approach is measuring real clustering and that the relatively small clusters observed for CTCF and Rad21 are not merely photo-blinking artefacts.

Third, we performed two-color labeling and imaging to unambiguously distinguish true clusters from photo-blinking. We labeled Halo-hCTCF in C32 U2OS cells with approximately equimolar con-centrations of PA-JF549 and PA-JF646 dyes and performed two-color PALM. Since each Halo-Tag can only bind one dye, any cluster composed of $N$ molecules should under ideal circumstances exhibit a binomial distribution of JF549 and JF646 molecules. That is, the probability of a cluster composed of $N$ CTCF molecules having $k$ JF549-conjugated CTCF molecules should follow:

$$P(X_{\mathrm{JF549}} = k) = \binom{N}{k} p_{\mathrm{JF549}}^{k} (1 - p_{\mathrm{JF549}})^{N-k}$$

where

$$p_{\mathrm{JF549}} = \frac{N_{\mathrm{JF549}}}{N_{\mathrm{JF549}} + N_{\mathrm{JF646}}}$$

is the fraction of all dye-labeled nuclear CTCF molecules that was labeled with JF549. Conversely, consider the other extreme case where all clusters are exclusively due to photo-blinking artifacts. In this extreme scenario, all apparent clusters should be exclusively composed of JF549-conjugated CTCF molecules or exclusively composed of JF646-conjugated CTCF molecules. If we plot the prob-ability density function for the fraction of JF549-labeled molecules in clusters, the idealized case should show a binomial distribution with a peak at $p_{\mathrm{JF549}}$. On the other hand, the extreme 'photo-blinking only' case should show a probability density function for the fraction of JF549-labeled mole-cules in clusters with peaks at 0 and 1 and nothing in between, corresponding to exclusive JF549 and exclusive JF646 clusters. Thus, to apply this analysis, we merged all JF549 and JF646 localiza-tions and applied the Bayesian cluster identification algorithm to the merged dataset. We then

analyzed all the called clusters that were composed of at least 10 detections. We consider only these clusters since for very small clusters the probability of finding clusters exclusively in one color is significant even in the ideal binomial case. In a given nucleus, hundreds of clusters fulfilled this criterion (>10 detections). To robustly compare this to the ideal binomial case, for each cluster of size $N$, we generated binomial random clusters using binornd in MATLAB. Finally, we compared the distribution of cluster compositions for the observed clusters and the binomial random clusters in *Figure 4—figure supplement 1E*. Since each nucleus had a slightly different fraction of molecules labeled with JF549 and JF646, we only show the distribution for a single nucleus. As can be seen, the deviation from the binomial case is small. Essentially, all clusters at this size contain molecules of both colors demonstrating that clustering is not exclusively a photo-blinking artifact. Thus, although some clustering is clearly due to photo-blinking, the majority of clusters are composed of multiple distinct molecules. To summarize the results for multiple cells, we also calculated the Kullback-Leibler divergence between the expected binomial and observed distributions for each cell. The mean Kullback-Leibler divergence was ~0.3 bits further demonstrating that most clusters are not a photo-blinking artifact. Finally, we note that a recent paper demonstrates that PA-JF549 shows limited photo-blinking (*Grimm et al., 2016*).

## Two-color dSTORM – data processing and pair cross correlation analysis

We processed two-color dSTORM data essentially identically to PALM data. After chromatic registration, blinking-correction and drift-correction using the same approach as for PALM analysis, nuclei were manually segmented using polygon segmentation based on a rough image generated by convolving the PSF with the single-molecule localizations and then blurring the image. We note that SNAP-tag dye-labeling is somewhat less specific than HaloTag labeling (*Figure 1—figure supplement 1*) – in particular, when we label wild-type cells that do not express a SNAP-tag protein with cp-JF549 (or any other SNAP dye) we observe enrichment along the nuclear envelope that does not disappear even after extensive washings. Labeling inside the nucleus, however, appears to be specific with cp-JF549, but less so with SNAP-TMR (compare *Figure 1—figure supplement 1B and C*). To avoid this affecting our dSTORM analysis, we segmented out the nuclear envelope during segmentation of the nucleus. Images (such as *Figure 4A*) were generated by binning single-molecule localizations into square pixel-bins of 10 nm and then false-color rendering JF549 localizations in green and JF646 localizations in magenta, such that saturating co-localization appears white. We note that co-localization of two single molecules are therefore not visible in these rendered images. Only overlap of clusters with saturating brightness appear white. Thus, most co-localizing CTCF and Rad21 molecules are not visible in *Figure 4A*. Thus, as a much more quantitative analysis we performed pair cross correlation analysis. Like pair correlation analysis, which quantifies the spatial interaction of proteins with themselves (i.e. clustering), pair cross correlation analysis quantifies spatial interactions between two different proteins. Thus, $C(r)$ quantifies enrichment between two different proteins as a function of interparticle distance, $r$. When the two proteins are independent (Complete Spatial Randomness (CSR)), $C(r)=1$ for all $r$. We calculate $C(r)$ using the whole nucleus and edge-correction as previously described (*Stone and Veatch, 2015*) using bins of 10 nm. The main way in which pair cross correlation can cause false-positive pair cross correlation is through fluorophore bleedthrough during simultaneous two-color imaging. E.g. if 561 nm excited J549 molecules emit enough far-red photons to be detected in the JF646 channel, this would result in high, but false-positive, pair cross correlation at small $r$. To rule out bleedthrough and any other bias, we also imaged a mES cell line stably expressing H2B-SNAP$_f$ transfected with a plasmid encoding a free Halo protein. We expect no significant co-localization between these proteins beyond mild exclusion from certain nuclear regions (e.g. nucleolar regions). In agreement, their experimentally observed pair cross correlation was not significantly different from CSR at any $r$. Since these cells were imaged under the same conditions as C59 Halo-mCTCF/mRad21-SNAP$_f$, this rules out the possibility that the observed pair cross correlation at small $r$ between CTCF and cohesin is due to fluorophore bleedthrough or any other technical artifact.

## Antibodies

Antibodies were as follows: ChromPure rabbit and mouse normal IgG from Jackson ImmunoResearch (West Grove, PA); anti-CTCF for Western Blot (WB) from Millipore (Temecula, CA) (EMD 07–

729), for ChIP and Co-IP from Abcam (ab128873); anti-Rad21 for WB and ChIP from Abcam (Cambridge, MA) (ab154769), for CoIP from Millipore (EMD 05–908); anti-SMC1 and anti-SMC3 from Bethyl (Montgomery, TX) (A300-055A, A300-060A); anti-FLAG from Sigma-Aldrich (F7425); anti-TBP, anti-H3, and anti-V5 from Abcam (ab51841, ab1791, ab9116).

## Chromatin immunoprecipitation (ChIP) and ChIP-seq libraries

ChIP assays in wild-type and double CTCF/Rad21 knock-in (clone C59) mouse JM8.N4 mES cells were performed essentially as described (*Testa et al., 2005*) with minor modifications. Cells were cross-linked for 5 min at room temperature with 1% formaldehyde-containing medium; cross-linking was stopped by PBS-glycine (0.125 M final). Cells were washed twice with ice-cold PBS, scraped, centrifuged for 10 min at 4000 rpm, resuspended in cell lysis buffer (5 mM PIPES, pH 8.0, 85 mM KCl, and 0.5% NP-40, 1 ml/15 cm plate) and incubated for 10 min on ice. During the incubation, the lysates were repeatedly pipetted up and down every 5 min. Lysates were then centrifuged for 10 min at 4000 rpm. Nuclear pellets were resuspended in six volumes of sonication buffer (50 mM Tris-HCl, pH 8.1, 10 mM EDTA, 0.1% SDS), incubated on ice for 10 min, and sonicated to obtain DNA fragments below 2000 bp in length (Covaris (Woburn, MA) S220 sonicator, 20% Duty factor, 200 cycles/burst, 100 peak incident power, 50 cycles of 30' on and 30' off). Sonicated lysates were cleared by centrifugation and 400–1600 µg of chromatin was diluted in RIPA buffer (10 mM Tris-HCl, pH 8.0, 1 mM EDTA, 0.5 mM EGTA, 1% Triton X-100, 0.1% SDS, 0.1% Na-deoxycholate, 140 mM NaCl) to a final concentration of 0.8 µg/µl, precleared with Protein A sepharose (GE Healthcare, Pittsburgh, PA) for 2 hr at 4°C and immunoprecipitated overnight with 8–16 µg of normal rabbit IgGs, anti-Rad21 or anti-CTCF antibodies. About 15% of the precleared chromatin was saved as input. Immunoprecipitated DNA was purified with the Qiagen (Germantown, MD) QIAquick PCR Purification Kit, eluted in 60 µl of water and analyzed by qPCR together with 2% of the input chromatin prior to ChIP-seq library preparation (SYBR Select Master Mix for CFX, ThermoFisher, see *Supplementary file 2* for primer sequences).

ChIP-seq libraries were prepared independently from two ChIP biological replicates using the Illumina (San Diego, CA) TruSeq DNA sample preparation kit according to manufacturer instructions with few modifications. We used 100 ng of ChIP input DNA (as measured by Fragment analyzer) and 50 µl of immunoprecipitated DNA as a starting material; Illumina adapters were diluted 1:50, and library samples were enriched through 18 cycles of PCR amplification. We assessed library quality and fragment size by qPCR and Fragment analyzer, and when necessary we performed an additional size selection step on agarose gel after PCR amplification to enrich for fragments between 150 and 500 bp. We sequenced four to eight multiplexed libraries per lane on the Illumina HiSeq4000 sequencing platform (single end-reads, 50 bp long) at the Vincent J. Coates Genomics Sequencing Laboratory at UC Berkeley, supported by NIH S10 OD018174 Instrumentation Grant.

## ChIP-seq analysis

Input, IgG, Rad21 and CTCF ChIP-seq raw reads from wild type and knock-in ESCs from two biological replicates (18 libraries total, see *Supplementary file 1*) were quality-checked with FastQC and aligned onto the mouse genome (mm10 assembly) using Bowtie (*Langmead et al., 2009*), allowing for two mismatches (-n 2) and no multiple alignments (-m 1). Enriched regions were visualized on the mm10 genome with the Integrative Genomics Viewer (IGV) (*Robinson et al., 2011*; *Thorvaldsdóttir et al., 2013*), after creating tiled data files from alignment files (igvtools count -w 50 -e 200). Peaks were called with MACS2 (–nomodel –extsize 250) (*Zhang et al., 2008*) combining inputs from the two replicates as a control, first for each biological replicate separately, and then, after having verified that results were highly reproducible, for the merged replicates (*Supplementary file 1*). Coverage and overlap between ChIP-seq peaks across samples and with previously published CTCF and Rad21 datasets were computed through Galaxy (*Blankenberg et al., 2010*; *Giardine et al., 2005*; *Goecks et al., 2010*), requiring a minimum 1 bp overlap between peak intervals (*Supplementary file 1*).

To create heatmaps, we used deepTools (version 2.4.1) (*Ramírez et al., 2016*). We first ran bamCoverage (–binSize 50 –normalizeTo1 $\times$ 2150570000 extendReads 250 –ignoreDuplicates -of bigwig) and normalized read numbers of WT and C59 IgG, CTCF and Rad21 merged replicates to 1x sequencing depth, obtaining read coverage per 50 bp bins across the whole genome (bigWig files).

We then used the bigWig files to compute read numbers across 6 kb centered on either WT CTCF or WT Rad21 peak summits as called by MACS2 (computeMatrix reference-point –reference-Point=TSS –upstream 3000 –downstream 3000 –missingDataAsZero –sortRegions=no). We sorted the output matrices by decreasing WT enrichment, calculated as the total number of reads within a MACS2 called ChIP-seq peak. Finally, heatmaps were created with the plotHeatmap tool (–average-TypeSummaryPlot=mean –colorMap='Blues' –sortRegions=no).

## RT-qPCR analysis

Total RNA was purified from cell pellets using RNeasy Plus Mini kit (Qiagen) and quantified by Nano-drop. For RT-qPCR, 1 µg of total RNA was retrotranscribed to cDNA with oligo(dT) primers (Ambion, Life Technologies, ThermoFisher) and Superscript III (Invitrogen, ThermoFisher). 2 µl of 1:40 cDNA dilutions were used for quantitative PCR (qPCR) with SYBR Select Master Mix for CFX (Applied Biosystems, ThermoFisher) on a BIO-RAD CFX Real-time PCR system (see *Supplementary file 2* for primer sequences).

## Western blot and co-immunoprecipitation (Co-IP) assays

Cells were collected by scraping from plates in ice-cold phosphate-buffered saline (PBS), pelleted, and flash-frozen in liquid nitrogen.

For Western blot analysis, cell pellets where thawed on ice, resuspended to 1 mL/10 cm plate of low-salt lysis buffer (0.1 M NaCl, 25 mM HEPES, 1 mM $MgCl_2$, 0.2 mM EDTA, 0.5% NP-40 and protease inhibitors), with 125 U/mL of benzonase (Novagen, EMD Millipore), passed through a 25G needle, rocked at 4°C for 1 hr and a NaCl solution was added to reach a final concentration of 0.2 M. Lysates were then rocked at 4°C for 30 min and centrifuged at maximum speed at 4°C. Supernatants were quantified by Bradford. Between 15 and 60 µg of proteins were loaded onto 9% Bis-Tris SDS-PAGE gel, transferred onto nitrocellulose membrane (Amershan Protran 0.45 um NC, GE Healthcare) for 2 hr at 100V, blocked in TBS-Tween with 10% milk for at least 1 hr at room temperature and blotted overnight at 4°C with primary antibodies in TBS-T with 5% milk. HRP-conjugated secondary antibodies were diluted 1:5000 in TBS-T with 5% milk and incubated at room temperature for an hour.

For Co-IP experiments, cell pellets where thawed on ice, resuspended to 1 ml/10 cm plate of cell lysis buffer (5 mM PIPES pH 8.0, 85 mM KCl, 0.5% NP-40 and protease inhibitors), and incubated on ice for 10 min. Nuclei were pelleted in a tabletop centrifuge at 4°C, at 4000 rpm for 10 min, and resuspended to 0.5 mL/10 cm plate of low salt lysis buffer with benzonase as above. For each sample, 1 mg of proteins was diluted in 1 mL of Co-IP buffer (0.2 M NaCl, 25 mM Hepes, 1 mM $MgCl_2$, 0.2 mM EDTA, 0.5% NP-40 and protease inhibitors), pre-cleared for 2 hr at 4°C with protein-G-sepharose beads (GE Healthcare Life Sciences) before overnight immunoprecipitation with 4 µg of either normal serum IgGs or specific antibodies as listed above. Some pre-cleared lysate was kept at 4°C overnight as input. Protein-G-sepharose beads precleared overnight in CoIP buffer with 0.5% BSA were then added to the samples and incubated at 4°C for 2 hr. After extensive washes in Co-IP buffer, proteins were eluted from the beads by boiling for 5 min in 2X SDS-loading buffer and analyzed by SDS-PAGE and Western blot.

## Datasets and accession numbers

The ChIP-seq data discussed in this publication have been deposited in NCBI's Gene Expression Omnibus (*Edgar et al., 2002*) and are accessible through GEO Series accession number GSE90994. We compared our ChIP-seq to previous ChIP-Seq studies of Rad21 and CTCF: (*Handoko et al., 2011*; *Nitzsche et al., 2011*; *Shen et al., 2012*) and GSE29218.

## Fluorescence recovery after photobleaching (FRAP) imaging

FRAP was performed on an inverted Zeiss (Germany) LSM 710 AxioObserver confocal microscope equipped with a motorized stage, a full incubation chamber maintaining 37°C/5% $CO_2$, a heated stage, an X-Cite 120 illumination source as well as several laser lines (only the 561 nm laser was used here). Images were acquired on a 40x Plan NeoFluar NA1.3 oil-immersion objective at a zoom corresponding to a 100 nm x 100 nm pixel size and the microscope controlled using the Zeiss Zen software. In most FRAP experiments, except where otherwise noted, 300 frames were acquired at either

one frame per second allowing 20 frames to be acquired before the bleach pulse to accurately estimate baseline fluorescence or 330 frames at one frame per 2 s again allowing 20 frames to be acquired before the bleach pulse. A circular bleach spot (r = 10 pixels) was chosen in a region of homogenous fluorescence at a position at least 1 μm from nuclear or nucleolar boundaries. The spot was bleached using maximal laser intensity and pixel dwell time corresponding to a total bleach time of ~1 s. We note that because the bleach duration was relatively long compared to the timescale of molecular diffusion, it is not possible to accurately estimate the bound and free fractions from our FRAP curves.

We generally collected data from 6 to 10 cells per cell line per condition per day, and all presented data are from at least three independent replicates on different days. To quantify and driftcorrect the FRAP movies (cell movement is an issue, especially for mES cells), we custom-wrote a pipeline in MATLAB. Briefly, we manually identify the bleach spot. The nucleus is automatically identified by thresholding images after Gaussian smoothing and hole-filling (to avoid the bleach spot as being identified as not belonging to the nucleus). We use an exponentially decaying (from 100% to ~85% of initial over one movie) threshold to account for whole-nucleus photobleaching during the time-lapse acquisition. Next, we quantify the bleach spot signal as the mean intensity of a slightly smaller circle (r = 0.6 μm), which is more robust to lateral drift. The FRAP signal is corrected for photobleaching using the measured reduction in total nuclear fluorescence (~15% over 300–330 frames at the low laser intensity used after bleaching) and internally normalized to its mean value during the 20 frames before bleaching. We correct for drift by manually updating a drift vector quantifying cell movement during the experiment. Finally, drift- and photobleaching corrected FRAP curves from each single cell were averaged to generate a mean FRAP recovery. We used the mean FRAP recovery in all figures and for model-fitting.

Model selection is a crucial step in FRAP experiments and has been studied extensively (*Mueller et al., 2008*, *2010*; *Sprague et al., 2004*). A full FRAP model considers both diffusion, the shape of the bleach spot and reactions (e.g. binding and unbinding). However, Sprague *et al.* identified circumstances under which simpler models are applicable (*Sprague et al., 2004*). Importantly, minimizing the number of fitted parameters is desirable because FRAP modeling tends to otherwise be prone to overfitting. Sprague *et al.* showed that when:

$$\frac{k_{ON}^* w^2}{D_{FREE}} \ll 1 \text{ and } \frac{k_{OFF}}{k_{ON}^*} \lesssim 1$$

Then a 'reaction dominant' FRAP model is most appropriate (*w* is the radius of the bleach spot). In the case of the second condition, for CTCF in both mES and U2OS cells, $k_{OFF} \approx k_{ON}^*$. Likewise, for mRad21-Halo in mESCs $k_{OFF} \approx k_{ON}^*$. Thus, the second condition suggests a reaction dominant model. For the first condition, we find:

Halo-mCTCF in mESCs: $\frac{k_{ON}^* w^2}{D_{FREE}} = \frac{0.015 s^{-1} \cdot (0.6\ \mu m)^2}{2.5\ \mu m^2 s^{-1}} = 0.0022 \ll 1$

mRad21 in mESCs (G1 phase): $\frac{k_{ON}^* w^2}{D_{FREE}} = \frac{0.0005 s^{-1} \cdot (0.6\ \mu m)^2}{1.5\ \mu m^2 s^{-1}} = 0.00012 \ll 1$

Thus, both CTCF and Rad21 lie within the reaction dominant parameter space and a reaction-dominant FRAP model is therefore the most appropriate choice. As has been demonstrated previously (*Sprague et al., 2004*), in the reaction-dominant parameter range, the FRAP recovery depends only on $k_{OFF}$ and we fit the FRAP recoveries to the reaction-dominant model below:

$$FRAP(t) = 1 - Ae^{-k_a t} - Be^{-k_b t}$$

After model-fitting (*Figure 2—figure supplements 2D* and *3C*), we used the slower off rate to estimate the residence time according to $\tau_s = \frac{1}{k_{off}}$.

In FRAP modeling, an important question is whether or not it is justifiable to ignore diffusion (as the above model does) and the radial shape of the bleach spot. Mueller *et al.* previously showed that ignoring diffusion can lead to serious errors for typical transcription factors which show rapid FRAP recovery (in the seconds to tens of seconds range) (*Mueller et al., 2008*). To test whether diffusion must be taken into account we plotted the radial shape of the bleach spot as a function of time. In general, if recovery is due to binding, the recovery should be mostly uniform across the bleach area, since all binding sites are equally likely to be sampled. If on the other hand diffusion dominates the recovery, the outer edges of the circle will recover first and the center of the circle

last, since unbleached molecules are diffusing in from the outside. As can be seen (*Figure 2—figure supplement 3E*), the radial profile of the bleach spot is flat and thus diffusion can be ignored in the FRAP modeling. We note that in previous studies on typical transcription factors, complete or near-complete FRAP recovery was generally observed in the 10–20 s range and here diffusion is critical (*Mazza et al., 2012*; *Mueller et al., 2008*; *Sprague et al., 2004*). But in the case of CTCF and cohesin, FRAP recovery is about two orders of magnitude slower, and thus, it is not surprising that diffusion can be ignored. Finally, Mueller *et al.* modeled the shape of the bleach spot as a Gaussian (*Mueller et al., 2008*), but showed that if the flat part of the bleach spot is used instead, equivalent results are obtained. Thus, in our case, we bleach a circle with a 1 μm radius but use a circle with a 0.6 μm radius to calculate the FRAP recovery, which is in the uniform area of the radial bleach profile. In addition to being equivalent to the full Gaussian description of the radial bleach profile, it has the advantage of being much more robust to cell drift, which is extensive for mES cells over the 11 min that most of our FRAP experiments last.

Finally, it came to our attention that during extended FRAP experiments (in the multi hour range), incomplete washout of Halo- or SNAP-dye can lead to artifactual FRAP recovery (*Rhodes et al., 2017*). This is most likely through dye binding to new protein produced after the bleach pulse. This can be corrected for by adding an excess of 'dark' Halo- or SNAP-ligand, such that any newly synthesized protein binds the dark ligand. However, this is unlikely to contribute significantly to FRAP recoveries on the minute timescale since we estimate that only around 1% of the total protein is replenished during our longest FRAP experiments. Consistently, we could not detect a difference in FRAP recovery after adding excess dark ligand (*Figure 2—figure supplement 3F*). We conclude that our FRAP experiments were not affected by this.

## Acknowledgements

We thank Luke Lavis for generously providing JF dyes, Gina M Dailey for extensive assistance with cloning, Astou Tangara for microscopy assembly and maintenance, and Dr. Kartoosh Heydari at the Li Ka Shing Facility for flow cytometry assistance. We thank Sheila Teves and other members of the Tjian and Darzacq labs, Douglas Koshland, Miriam Huntley, James Rhodes and Kim Nasmyth, and Leonid Mirny and other 4D Nucleome consortium members for insightful comments on the manuscript. This work was performed in part at the CRL Molecular Imaging Center, supported by the Gordon and Betty Moore Foundation. This work used the Vincent J Coates Genomics Sequencing Laboratory at UC Berkeley, supported by NIH S10 Instrumentation Grants 10RR029668 and S10RR027303. ASH is a postdoctoral fellow of the Siebel Stem Cell Institute. This work was supported by NIH grants UO1-EB021236 and U54-DK107980 (XD), the California Institute of Regenerative Medicine grant LA1-08013 (XD), and by the Howard Hughes Medical Institute (003061, RT). ChIP-Seq data has been deposited at NCBI GEO under accession code GSE90994. A preprint describing this work was first available on BioRxiv December 2016: http://www.biorxiv.org/content/early/2016/12/13/093476

## Additional information

### Competing interests

RT: President of the Howard Hughes Medical Institute (2009-present), one of the three founding funders of eLife, and a member of eLife's Board of Directors. The other authors declare that no competing interests exist.

### Funding

| Funder | Grant reference number | Author |
| --- | --- | --- |
| Siebel Stem Cell Institute | | Anders S Hansen |
| Howard Hughes Medical Institute | 003061 | Robert Tjian |
| California Institute of Regenerative Medicine | LA1-08013 | Xavier Darzacq |

| National Institutes of Health | UO1-EB021236 | Xavier Darzacq |
| National Institutes of Health | U54-DK107980 | Xavier Darzacq |

The funders had no role in study design, data collection and interpretation, or the decision to submit the work for publication.

### Author contributions

ASH, Conceptualization; design of experiments; Performed genome-editing of cell lines, conducted all imaging experiments, developed mathematical models, wrote code and analyzed the data; Writing-original draft; Writing-review and editing; IP, Performed co-IP, ChIP-Seq and cell line characterization; CC, Performed co-IP, ChIP-Seq and cell line characterization. Writing-review and editing; RT, XD, Conceptualization; Supervision; Writing-review and editing

### Author ORCIDs

Anders S Hansen, http://orcid.org/0000-0001-7540-7858
Robert Tjian, http://orcid.org/0000-0003-0539-8217
Xavier Darzacq, http://orcid.org/0000-0003-2537-8395

# Additional files

### Supplementary files

• Supplementary file 1. Table with ChIP-Seq relevant information.

• Supplementary file 2. Supplementary information and table with primer sequences.

### Major datasets

The following dataset was generated:

| Author(s) | Year | Dataset title | Dataset URL | Database, license, and accessibility information |
|---|---|---|---|---|
| Sejr Hansen A, Cattoglio C, Pustova I, Tjian R, Darzacq X | 2017 | Nuclear organization and dynamics of CTCF and cohesin | https://www.ncbi.nlm.nih.gov/geo/query/acc.cgi?acc=GSE90994 | Publicly available at the NCBI Gene Expression Omnibus (accession no: GSE90994) |

The following previously published datasets were used:

| Author(s) | Year | Dataset title | Dataset URL | Database, license, and accessibility information |
|---|---|---|---|---|
| Nitzsche A, Paszkowski-Rogacz M | 2011 | The Cohesin Complex Cooperates with Pluripotency Transcription Factors in the Maintenance of Embryonic Stem Cell Identity | https://www.ncbi.nlm.nih.gov/geo/query/acc.cgi?acc=GSE24030 | Publicly available at the NCBI Gene Expression Omnibus (accession no: GSE24030) |
| Handoko L, Xu H, Li G, Ruan Y, Wei C | 2011 | CTCF-Mediated Functional Chromatin Interactome in Pluripotent Cells | https://www.ncbi.nlm.nih.gov/geo/query/acc.cgi?acc=GSE28247 | Publicly available at the NCBI Gene Expression Omnibus (accession no: GSE28247) |
| Shen Y, Yue F, Ren B | 2012 | A draft map of cis-regulatory sequences in the mouse genome [ChIP-Seq] | https://www.ncbi.nlm.nih.gov/geo/query/acc.cgi?acc=GSE29218 | Publicly available at the NCBI Gene Expression Omnibus (accession no: GSE29218) |

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

## Appendix 1

### Estimation of the fraction of CTCF and cohesin molecules involved in looping

Since both CTCF and cohesin have functions beyond regulating chromatin looping, an important question is which fraction of chromatin-bound CTCF and cohesin sites are involved in chromatin looping. Conventionally, the number of occupied binding sites are assessed using ChIP-Seq and identified as peaks significantly above a background threshold. Experimentally, a spectrum of binding enrichments is always observed and peak calling involves a somewhat arbitrary discretization step. Using MACS2 (*Zhang et al., 2008*) and standard parameters (Materials and methods), we call 68,077 CTCF ChIP-Seq peaks in wild-type mESCs and a similar number in Halo-mCTCF knock-in cells (C59; see *Supplementary file 1* for full details). Likewise, for cohesin we observe 33,434 ChIP-Seq peaks of which 97% of the peaks overlap with a CTCF peaks. Thus, the cohesin peaks appear to be a subset of CTCF peaks and there appears to be significant cohesin binding at many other CTCF peaks, albeit below the peak-calling threshold.

What fraction of CTCF/cohesin sites are involved in looping? As for calling peaks using ChIP-Seq data, loops are also generally called by thresholding Hi-C data and appear as corner-peaks in the Hi-C interaction matrix. Different groups have used different thresholds and Hi-C data at different resolutions and accordingly have reported different numbers of loops (*Jin et al., 2013*; *Rao et al., 2014*; *Sanyal et al., 2012*). The highest resolution Hi-C data published to date is from Rao *et al.* and they report ~10,000 loops using a very stringent and conservative loop-calling algorithm in GM12878 cells (*Rao et al., 2014*). The same group called substantially fewer loops in other cell lines sequenced at a lower sequencing depth (lower resolution Hi-C). However, using a method called Aggregate Peak Analysis (APA), which allows Hi-C maps at different resolutions to be compared, Rao *et al.* found that the fewer loops were due to the lower sequencing depth rather than an absence of loops in these cell lines. In fact, they found that loops were largely conserved between different cell lines and between human and mouse cells. Thus, it seems like the ability to call loops depends on sequencing depth and thus, it seems likely that in the future when even higher resolution Hi-C data may be available, the number of high-confidence loops will significantly exceed 10,000. According to Rao *et al.*, almost all Hi-C loops are anchored by both CTCF and cohesin. Thus, a lower bound estimate would be that ~20,000 CTCF and Cohesin ChIP-Seq sites anchor loops. However, as also pointed out by Rao *et al.* and clearly illustrated in *Figure 2* of an informative recent review by Merkenschlager and Nora (*Merkenschlager and Nora, 2016*), many loops appear to be anchored by clusters of CTCF/cohesin binding sites. Thus, since multiple CTCF and cohesin ChIP-Seq sites can anchor the same loop, 20,000 seems to be too low a bound. If we further take into account that future Hi-C studies, which achieve even greater resolution, will likely call even more loops, it seems reasonably conservative to take ~25,000 CTCF and cohesin ChIP-Seq peaks as the number of peaks involved in looping. While this is clearly a rough and somewhat speculative estimate, if we compare this to the MACS2-called ChIP-seq peaks we find that ~25,000/68,077 or ~37% of CTCF ChIP-Seq called binding sites and ~25,000/33,434 or ~75% of cohesin ChIP-Seq called binding sites are involved in chromatin looping. In the main text of the manuscript, we refer to this as around one-third of CTCF sites and as a majority of cohesin sites. We also note that within the extrusion model, a significant fraction of cohesin molecules that are topologically engaged on chromatin may be actively travelling across the chromosome (i.e. 'extruding') and this fraction is unlikely to be picked up by any ChIP-Seq peak-calling analysis. This fraction would appear indistinguishable from cohesin molecules bound at specific loop boundaries in our FRAP analysis. Nevertheless, among cohesin molecules that remain at a specific location for an extended period, i.e. the fraction likely to result in ChIP-Seq peaks, the majority appears around loop boundaries.

For CTCF sites, we would also like to note that the CTCF sites involved in looping tend to be the ones with the highest ChIP-Seq enrichment (*Merkenschlager and Nora, 2016*). The ChIP-Seq enrichment should be approximately proportional to the fraction of time the binding site is occupied. Thus, the CTCF sites that make up loop anchors are likely bound a higher fraction of the time than other CTCF sites. This is important, because the probability of observing CTCF binding to a particular site in our imaging experiments should also scale with the fractional occupancy of this site. Thus, in our single-molecule tracking experiments (*Figure 2A–D*), we are over-sampling precisely the CTCF binding events at loop anchors. Thus, most likely, of the binding events that we observe in *Figure 2A–D*,>37% are involved in looping. Further support for this interpretation, comes from the observation that overexpressing CTCF greatly increases the rate of FRAP recovery (*Figure 2—figure supplement 2B*: black curve vs. red and blue curves). The simplest explanation for this over-expression artefact is that when the abundance of CTCF substantially increases, many CTCF molecules now start binding 'poor' CTCF sites on chromatin and accordingly the apparent residence time is decreased. For these reasons, we believe that our estimate that around one-third of CTCF sites are involved in looping is a very conservative estimate and we believe that this is a lower bound.

In the case of cohesin, cohesin clearly has many other functions besides looping such as sister chromatids cohesion and DNA repair through homologous recombination. However, most of these functions only exist from S-phase to division during the cell cycle. Thus, our estimate that a majority of cohesin molecules are involved in chromatin looping apply to G1-phase, where sister chromatid cohesion and homologous recombination does not occur.

Moreover, we note that both ChIP-Seq and Hi-C and the other 3C variants (e.g. 4C and 5C) all provides snapshots of large cell populations. Thus, a ChIP-Seq peak and a Hi-C loop shows that a binding site is occupied and that a loop exists, in a fraction of cells, but it is extremely difficult to estimate how big this fraction is from these techniques. And even with DNA FISH measurements, it can be difficult to ascertain precisely the frequency with which a loop occurs in a cells (*Fudenberg and Imakaev, 2016*). A very recent paper used single-cell Hi-C to estimate that loops form in 62.1% of mouse ES cells (*Stevens et al., 2017*). This is a somewhat higher estimate than what most DNA-FISH studies find. Nevertheless, if we assume that the fractional occupancy of CTCF sites is significantly less than 62.1%, which is likely the case (but cannot be determined with knowing the absolute number of CTCF molecules per cell), this would also imply that a much higher fraction than 37% of CTCF binding sites is involved in looping. However, because we do not yet have good data on the fractional binding site occupancy and on the exact number and frequency of loops, it is difficult to say with certainty what fraction of CTCF molecules are truly involved in looping.

Finally, we note that if loops are formed by a cohesin-mediated extrusion mechanism (*Fudenberg et al., 2016*; *Sanborn et al., 2015*), many cohesin molecules will be actively extruding loop and thus involved in looping, but not actually show up in ChIP-Seq as a peak. This is because for the extrusion model to work, cohesin has to extrude quite quickly along chromatin and thus its occupancy is effectively 'spread out' and will not show up in a ChIP-Seq experiment as a peak and thus will not be called. This may be one reason, why we find more CTCF ChIP-Seq peaks than cohesin peaks. Thus, it is very plausible that more cohesin than CTCF molecules will be chromatin associated even though fewer cohesin ChIP-Seq peaks are called.

