## [Decision Letter]

Thank you for submitting your article "CTCF and Cohesin Regulate Chromatin Loop Stability with Distinct Dynamics" for consideration by *eLife*. Your article has been reviewed by three peer reviewers, one of whom is a member of our Board of Reviewing editors, and the evaluation has been overseen by Kevin Struhl as the Senior Editor. The following individual involved in review of your submission has agreed to reveal his identity: Lothar Schermelleh (Reviewer #2).

Your manuscript has been assessed by three reviewers and the consensus is that the material is potentially suitable for publication in *eLife*. Nevertheless, there are a number of significant and minor issues that you should take into account in revising your manuscript and we propose you undertake some further experimentation and analysis. Below, the overall view, comments and concerns are listed.

The work provides a thorough and extensive study of the in vivo dynamics and differential binding kinetics of CTCF and Cohesin to chromosomes in mouse ES and human U2OS cells using advanced single molecule and FRAP imaging as well as complementary CoIP and ChIP-seq experiments. The experimental approaches (ranging from the tagging of the endogenous proteins by CRISPR/Cas9 to the careful modelling of single molecule experiments) and quantitative analyses are of highest standard, and the technical execution, data quality and strength of the conclusions in this manuscript are sound for the most part; nevertheless, see below.

This work presents compelling evidence for the dynamic nature of "loop maintenance complexes" required to stabilize topological chromatin domains in interphase. By providing novel high quality quantitative data (resolving some inconsistencies in previous literature), the work contributes significantly to the understanding of how cohesin and CTCF cooperatively organize chromatin topology. The authors show that the kleisin [Rad21] subunit of cohesin [a surrogate for cohesin complexes] displays significantly longer binding to DNA (timescale of ~30 min) than CTCF (timescale of ~1 min), leading to the hypothesis that cohesin and CTCF form dynamic LMC complexes in which the CTCF molecules exchange rapidly. Furthermore, the authors show that also the search strategies of the two chromatin binders are different with CTCF rapidly identifying its target-sites, and cohesin binding about 30 times more infrequently. It is proposed [and this is not novel] that cohesin's ability to maintain two duplex segments within its proteinaceous ring has a key role in structuring the loops at CTCF binding sites [thereby forming TADs]

Major comments

1) Binding kinetics of Cohesin/CTCF at loop sites

The long-lived binding of CTCF [~1 min] and cohesin [~20 min] extracted by FRAP and SMT are averaged over both "loop-sites" and other chromosomal regions. Therefore, there is the possibility that only a small fraction of the observed CTCF-bound population is engaged in stable binding at chromatin loops. Furthermore, stably-bound cohesins may be present at both CTCF-bound sites and at other positions on chromosomes. Relevant to this point, comparison of the ChIP-seq data with published HiC data (Appendix 1) seem to indicate that cohesin binding sites are mostly on loops, while for CTCF about 30% of the ChiP seq peaks are found in loops. As the authors point out these calculations are rather rough because calling ChiP peaks depends on the selection of thresholds. Further it is not clear how SMT data and ChIP relate: would all the measured residence times from few seconds to several minutes show-up as a peak in the ChIP data, or only the most stable ones? With these uncertainties, it is hard to understand what is the sub-population of single CTCF molecules that is potentially engaged with cohesin at loop sites [and vice versa]. Is it possible that only a small fraction of very-long lived CTCF binding events are those involved in chromatin looping? If the authors think that they can exclude this possibility from their current data, they should discuss this issue more incisively. Otherwise, to provide definitive evidence that CTCF molecules engaged in LMCs are transiently interacting with chromatin the authors could compare CTCF binding at regions enriched and devoid of cohesin (Ideally by single molecule imaging on CTCF followed by PALM on cohesin on the same cell).

2) Modelling of SMT data

The method devised by the authors to quantify bound fractions is a nice and justified simplification of a previously described mathematical model to extract bound fractions from distribution of displacements, based on acquiring movies at a very fast time rate (>200Hz) during SMT. Nevertheless, one reviewer has serious concerns on the choice of limiting the analysis to the first 7 displacements of each track, even for molecules residing in the observation slice much longer, since this could result in a severe underestimation of the bound fraction. In this reviewer's opinion, Monte-Carlo simulations showing that this is not the case are needed. The reviewer explains their concern with a simplified example: consider the situation in which we have 20 molecules in one nucleus and a bound fraction of 50%. By definition this mean that at each frame half of the detected molecules will belong to "bound segments" and half to free ones. Let's also consider an observation slice that spans 1/10 of the thickness of the nucleus, so that per-frame there will be one molecule free and one molecule bound. Now as it is unlikely that a molecule will diffuse out of the detection volume with the very fast tracking rate between one frame and the next (probability to diffuse out roughly equal to e^(-deltaZ^2/4*D*δ_t)), but it gets more likely as deltaT time increases. The correction for the observation volume accounts for this, being a small correction when deltaT is small, and getting bigger and bigger for longer deltaTs. Conversely, discarding the segments of the tracks after the 7th one, is a hard threshold that similarly affects data at short intervals and data at long intervals. At the shorter time-interval, if we consider, for example an acquisition of 100 frames, the bound track will always be the same (as residence times are very long in the experimental situation of the paper), and therefore will contribute to the histogram of displacements with just 7 jumps (all the others are discarded). For the free molecules, instead the situation will be radically different. Each of them will be visible for a certain number of frames that on their diffusion coefficient, and then get out of the observation volume. The point is that when a molecule gets out of focus, it will be substituted – on average – by another one, that will be counted as a new one (each still contributing 7 displacements). By a back of the envelope calculation, if each free molecule stays within the detection volume for n frames on average, we would count a number of displacements associated to the free molecules equal to: min(n-1, 7)x(100/n).

For example, for the case in which n = 7, we would count 84 displacements associated to the free molecules, compared to the 7 jumps associated to the bound molecule. Therefore the estimated bound fraction would be roughly (7/(7+84)) = 8%, strongly underestimated compared to the 50% that we have imposed at the beginning of this calculation. This problem should not be there, instead, if bound and free molecules last for the same amount of time within the observation volume, but, I guess that this is not the case in the experimental settings, otherwise no correction for the limited thickness of the detection volume would be necessary.

Furthermore, the estimation of the bound fraction and the estimation of the residence times are performed on two very different timescales (4.5-30 ms vs. >3s). It would be nice if the authors could exclude the presence of a fast dissociating population (in the gap between the two measurements), that might affect the calculation of the search times. Maybe by showing that the bound fraction does not change when acquiring data at a slower frame time (in the 50-200 ms range)?

3) FRAP and modelling of FRAP.

The details of the choice of the model for FRAP data are lacking. A double exponential assumes that diffusion is much faster than the unbinding process and the acquisition rate, and therefore does not contribute to recovery. This assumption needs to be tested/justified.

While it seems reasonable that most of the molecules will have diffused out from the bleach region in the time between two acquisitions (75%-80% given the diffusion coefficients reported by the authors), diffusion might still contribute to the first time-points of the recovery curve. The authors might test for the role of diffusion by plotting the evolution of the bleach profile over time, which should not broaden, if the assumption of the negligible role of diffusion is correct. Also, the estimations for CTCF residence time are quite different for SMT and FRAP (1 min vs. 3-4 min): the authors indicate that this might be due to multiple binding and unbinding events during the recovery. I have some doubts that in this case a multi-exponent fitting of the recovery curve is the most appropriate model. More importantly this interpretation is unlikely as the SMT data seem to indicate that the molecules spend about 60s between two stable binding events, and of these about 30s are spent diffusing freely in the nucleus. In these 30s the molecule should diffuse far away from the bleaching region, so it is unclear how the multiple binding events could significantly contribute to the recovery. One possibility is anomalous diffusion, that would constrain the spread of CTCF molecules after unbinding. Do the authors have any evidence from the SMT data for anomalous behavior of free CTCF molecules? Could relabelling by free dye during recovery [as reported by others] impact on the interpretation of the FRAP?

4) Two color dSTORM analysis

(Subsection “CTCF and cohesin co-localize in cells and show a clustered nuclear organization”, Figure 4 and Figure 4—figure supplement 1).

The two-color dSTORM image (Figure 4) shows more Rad21 (localization) signals than CTCF signals. This seems not in line with the ChIP-seq. peak numbers (68.000 CTCF, 33.400 Rad21, 97% overlap of Rad21 peaks with CTCF) that would lead to expect this to be the other way round. The authors should offer an explanation for this apparent inconsistency.

Furthermore, there seems to be a prominent enrichment of Rad21 along the nuclear envelope, which I have not been aware from other paper describing the nuclear distribution of cohesin using conventional imaging. Hence it may be useful to compare the dSTORM images with widefield deconvolution or confocal images of the same cell line(s).

For the additional clustering analyses (Figure 4—figure supplement 1) the authors use a 2D PALM localization approach. With the depth of the optical slice being around 700 nm, could this have potential effects on the clustering analyses? How should the clustering be interpreted? Do the clusters coincide with the co-localizing regions in the dSTORM images? Are the clusters equivalent to the LMCs, whereas non-clustering molecules are rather belonging to the free fraction? If 25% of CTCF or 35% of Rad21 molecules are in clusters, then only a subset of LMCs are in clusters, right? Then again, if 15% NLS-Halo is found in clusters, how relevant is the slight increase cluster fraction for CTCF and Rad21? Subsection “CTCF and cohesin exhibit distinct nuclear search mechanisms” and Table 1: In the text 3 fractions are distinguished (bound to cognate site; non-specifically associated with chromatin; free 3D diffusion), whereas in the table only two fractions are distinguished (bound total vs. bound specific). Why not listing the 3 fractions as in the text, to avoid confusion? Could the "non-specific binding" or "1D sliding" not also explained by corralled or anomalous diffusion? Does the Rad21-HaloTag kinetic data imply that Rad21 (and other Cohesin subunits) are predominantly present as either free (searching) or bound complexes, but not as single proteins?

[Editors' note: further revisions were requested prior to acceptance, as described below.]

Thank you for resubmitting your work entitled "CTCF and Cohesin Regulate Chromatin Loop Stability with Distinct Dynamics" for further consideration at *eLife*. Your revised article has been favorably evaluated by Kevin Struhl (Senior editor), a Reviewing editor, and one reviewer.

The manuscript has been substantially improved but there are some remaining issues that need to be addressed before acceptance, as outlined below:

Below, we append, the reviewer's comments, that we invite you to respond to, and which will help you make your final revisions.

Reviewer #3:

In the revised version of the manuscript, Hansen et al. have made substantial efforts to answer the questions raised, by performing computer simulations and restating parts of the text. I am a little disappointed that despite some additional "wet" experiments were requested, the authors did not perform any. Nevertheless their arguments against my criticism are now more convincing, and I support publication in *eLife*.

The authors performed computer simulations to prove that limiting the population of the distribution of displacements to the first 7 frames of each track, however I see some problems with these simulations.

For the paper purpose, my suggestion would be to drop the description of these simulations from the text: they have a better argument, given by the fact that whether they analyze their experimental data by looking at all of the jumps, or only at the first 7 they essentially obtain the same results.

---

## [Author Response]

*Major comments*

*1) Binding kinetics of Cohesin/CTCF at loop sites*

*The long-lived binding of CTCF [~1 min] and cohesin [~20 min] extracted by FRAP and SMT are averaged over both "loop-sites" and other chromosomal regions. Therefore, there is the possibility that only a small fraction of the observed CTCF-bound population is engaged in stable binding at chromatin loops. Furthermore, stably-bound cohesins may be present at both CTCF-bound sites and at other positions on chromosomes. Relevant to this point, comparison of the ChIP-seq data with published HiC data (Appendix 1) seem to indicate that cohesin binding sites are mostly on loops, while for CTCF about 30% of the ChiP seq peaks are found in loops. As the authors point out these calculations are rather rough because calling ChiP peaks depends on the selection of thresholds. Further it is not clear how SMT data and ChIP relate: would all the measured residence times from few seconds to several minutes show-up as a peak in the ChIP data, or only the most stable ones? With these uncertainties, it is hard to understand what is the sub-population of single CTCF molecules that is potentially engaged with cohesin at loop sites [and vice versa]. Is it possible that only a small fraction of very-long lived CTCF binding events are those involved in chromatin looping? If the authors think that they can exclude this possibility from their current data, they should discuss this issue more incisively. Otherwise, to provide definitive evidence that CTCF molecules engaged in LMCs are transiently interacting with chromatin the authors could compare CTCF binding at regions enriched and devoid of cohesin (Ideally by single molecule imaging on CTCF followed by PALM on cohesin on the same cell).*

This is a very important issue and the reviewers raise multiple points and we have attempted to address all of these points. First, based on available data, we can rule out that the loop-involved CTCF population is a tiny subpopulation. Second, we performed Monte Carlo simulations of CTCF binding and show that our SMT approach is quite sensitive to even very small subpopulations.

Regarding the first point, we do not believe that a very long-lived but tiny subset of “looping CTCF” events is escaping our notice. Here are our arguments for why this is the case:

In our SMT/FRAP experiments (Figure 2), we can readily distinguish specific binding from non-specific binding and we report a CTCF binding time of ~1-2 min at specific cognate binding sites.Transient binding at non-specific, non-cognate sites will not lead to sufficient enrichment to be called as a peak in ChIP-seq.Thus, the longer (~1-2 min) binding events in our imaging experiments correspond to binding to cognate sites and it is these binding sites that show up in ChIP-seq experiments.ChIP-seq enrichment should be proportional to the fraction of time the binding site is occupied.Likewise, the probability of seeing binding to a particular cognate site should also scale with the fraction of time it is occupied and thus its ChIP-seq enrichment.Therefore, in our imaging experiments, we are predominantly sampling the binding events at the cognate CTCF sites with the strongest enrichment.Thus, out of the 68,077 called CTCF ChIP-seq sites, we are predominantly seeing binding at the subset with the highest ChIP-seq enrichment.Around ~25,000/68,077 (Appendix 1) CTCF binding sites are involved in looping.The CTCF sites involved in looping tend to be the ones with the highest enrichment in ChIP-seq (Fudenberg et al., 2016; Merkenschlager and Nora, 2016).Therefore, our imaging experiments oversamples precisely the CTCF binding events involved in looping.Cognate CTCF sites with ChIP-seq peaks are not always occupied; at present, we do not know precisely the fractional occupancy but we estimate it to be in the 15-50% range (~60-200k molecules of CTCF per cell (from PALM estimate); 50% of CTCF bound to cognate sites, i.e. 30-100k; 3 haploid genomes in average cell (mid-S phase), i.e. ~204k sites;).A very recent single-cell Hi-C paper (Steven et al., 2017, 3D structures of individual mammalian genomes studied by single-cell Hi-C, Nature, 2017) found that contacts between CTCF/cohesin loop boundaries occur in 62.1% of mES cells.These numbers (15-50% fractional occupancy of average ChIP-seq site, but 62.1% contacts between loops boundaries) provide additional evidence that loop boundary sites (62%) are bound more often than average sites (15-50%).While some uncertainty is unavoidable, this would suggest that extremely conservatively ~1/3, but more likely >50% of all observed CTCF binding events occurred at loop boundaries.

Based on this argument, we exclude that only a tiny fraction of the observed CTCF binding events are actually involved in looping. Regardless, since we see essentially full mCTCF FRAP recovery in 11 min, we can say that even the most stable CTCF binding events are shorter than cohesin binding, thus it is impossible for loops to be stably held together by CTCF.

Secondly, we wanted to determine how small a sub-population of very stable binding we would be able to detect in our SMT imaging. We performed Monte Carlo simulations of CTCF binding using the Gillespie algorithm and the following assumptions:

Photobleaching is a Poisson process with a rate constant of 1/120 s.Non-specific binding is a Poisson process with a residence time of 1 s and 30% of all events are non-specific.A very stable subpopulation binds as a Poisson process with a residence time of 10 min and X% of all events are in this very stable pool.Specific binding is a Poisson process with a residence time of 1 min and 100% – 30% – X% of all events are specific.

We then simulated CTCF binding for 100,000 binding events using the Gillespie algorithm and plotted the 2-exponential fit (same as in manuscript) as a function of X, the fraction of very stable events. The results are shown in Figure 5:

Author response image 1.**DOI:**
http://dx.doi.org/10.7554/eLife.25776.024

As can clearly be seen, the presence of a very stable subpopulation causes the 2-exponential fit to fail. Even with a tiny sub-population of 2% with residence time of 10 min instead of 1 min, the double-exponential fit totally fails (we do least-squares fitting here as in the manuscript). Only at around 1% or less, do the very stable subpopulation result in a reasonable fit like we saw experimentally.

Thus, our Gillespie simulations suggest that our detection limit is about 1%. Based on the ChIP-seq/Hi-C comparison, at the very least ~35% of binding events are involved in looping. Thus, comparing these numbers (35-fold difference), we believe we can exclude the possibility that loops are maintained by a tiny sub-population of CTCF that is escaping detection in our SMT experiments (Figure 2).

We have therefore discussed this matter more incisively as suggested by the reviewers. We have updated and re-written our Discussion and we have changed the main text several places. We now say that:

“We note that a CTCF RT of ~1 min is a genomic average and that some binding sites likely exhibit a slightly longer or shorter mean residence time. We also note that we are likely oversampling binding events at the CTCF binding sites with the strongest ChIP-seq enrichment (Figure 1), which tends to be the sites involved in looping (Merkenschlager and Nora, 2016).”

“Although we cannot completely exclude a very small population (<5%) of CTCF or cohesin molecules with a somewhat shorter or longer RT, these RTs reflect chromatin-bound CTCF/cohesin. Since at least one-third of CTCF and the majority of G1 cohesin molecules bound to chromatin mediate looping (see Appendix 1 for estimate), we are confident that these RTs hold for most CTCF/cohesin molecules involved in looping.”

Regarding the 2-color SMT-followed-by-PALM experiment, it is a very challenging experiment due to significant cell drift and chromatin movement on the minute time-scale and beyond the scope of this paper.

*2) Modelling of SMT data*

*The method devised by the authors to quantify bound fractions is a nice and justified simplification of a previously described mathematical model to extract bound fractions from distribution of displacements, based on acquiring movies at a very fast time rate (>200Hz) during SMT. Nevertheless, one reviewer has serious concerns on the choice of limiting the analysis to the first 7 displacements of each track, even for molecules residing in the observation slice much longer, since this could result in a severe underestimation of the bound fraction. In this reviewer's opinion, Monte-Carlo simulations showing that this is not the case are needed. The reviewer explains their concern with a simplified example: consider the situation in which we have 20 molecules in one nucleus and a bound fraction of 50%. By definition this mean that at each frame half of the detected molecules will belong to "bound segments" and half to free ones. Let's also consider an observation slice that spans 1/10 of the thickness of the nucleus, so that per-frame there will be one molecule free and one molecule bound. Now as it is unlikely that a molecule will diffuse out of the detection volume with the very fast tracking rate between one frame and the next (probability to diffuse out roughly equal to e^(-deltaZ^2/4*D*δ_t)), but it gets more likely as deltaT time increases. The correction for the observation volume accounts for this, being a small correction when deltaT is small, and getting bigger and bigger for longer deltaTs. Conversely, discarding the segments of the tracks after the 7th one, is a hard threshold that similarly affects data at short intervals and data at long intervals. At the shorter time-interval, if we consider, for example an acquisition of 100 frames, the bound track will always be the same (as residence times are very long in the experimental situation of the paper), and therefore will contribute to the histogram of displacements with just 7 jumps (all the others are discarded). For the free molecules, instead the situation will be radically different. Each of them will be visible for a certain number of frames that on their diffusion coefficient, and then get out of the observation volume. The point is that when a molecule gets out of focus, it will be substituted – on average – by another one, that will be counted as a new one (each still contributing 7 displacements). By a back of the envelope calculation, if each free molecule stays within the detection volume for n frames on average, we would count a number of displacements associated to the free molecules equal to: min(n-1, 7)x(100/n).*

*For example, for the case in which n = 7, we would count 84 displacements associated to the free molecules, compared to the 7 jumps associated to the bound molecule. Therefore the estimated bound fraction would be roughly (7/(7+84)) = 8%, strongly underestimated compared to the 50% that we have imposed at the beginning of this calculation. This problem should not be there, instead, if bound and free molecules last for the same amount of time within the observation volume, but, I guess that this is not the case in the experimental settings, otherwise no correction for the limited thickness of the detection volume would be necessary.*

The reviewer raises an important point and we agree with the underlying problem that with infinitely photostable dyes, freely diffusing molecules outside the focal plane will be able to diffuse into focus and potentially lead to over-counting of the free fraction. We provide two answers: one “practical” using our raw data, and one “ideal” following the thought-example where photobleaching is very rare.

First, practically, it turns out that there is a very small effect of limiting the analysis to the first 7 displacements of each track on the estimated bound fraction. This is because the lifetime of the JF646 dye is very short under our fast-tracking conditions where we use high laser powers (in addition to photobleaching, we see significant blinking; after all, the principle of dSTORM is that high excitation power forces molecules into a blinking state; blinking does not bias towards either bound or free molecules). For example, the mean trajectory length for U2OS Halo-3xNLS is only 1.97 frames and the mean trajectory length for mESC C87 Halo-mCTCF is only 3.34 frames (due to differences in bound fractions and diffusion constants). Thus, in practice, the molecules that move out of focus generally bleach before they can re-appear (note that the axial HiLo-illumination slice is ~4µm, such that molecules can bleach outside of the axial detection window).

To show that there is no major bias because of photobleaching/blinking, we will use mESC C87 Halo-mCTCF 225 Hz SMT as an example. We took the full dataset (44 different cells across 5 different replicates; 20,000 frames per cell; 574,641 displacements in total). We then applied our SMT model to calculate the total fraction bound (specific+non-specific) as a function of the fraction of trajectories that were included ranging from the first 7 jumps to the entirety of all trajectories (a minor clarification: when we say the first 7 jumps, this includes 6 jumps of 1△τ, 5 jumps of 2△τ, and so on; we have now added this to the Materials and methods). As can be seen from the left plot in Figure 6, the effect is relatively minor (going from 68% to 75%).

Author response image 2.**DOI:**
http://dx.doi.org/10.7554/eLife.25776.025

However, in addition to the bias towards free molecules identified by the reviewer, there is another compensatory bias against free molecules. Namely, with molecules diffusing around the nucleus there will be frequent cases where the free molecule will enter the detection volume for a single frame and then diffuse out and bleach out-of-focus. These molecules are never counted, because in the absence of a jump/displacement (tracked for 2 consecutive frames), there is no contribution to the displacement histogram. As shown in the plot on the right in Figure 6 (Monte Carlo simulations of axial Brownian diffusion in and out of our detection volume), even for a slow-diffusing molecule like CTCF with D ~ 2.5 µm2/s, around 20% of all molecules move out-of-focus before the second localization and thus are not counted. Thus, in the practical case of our data where the lifetime of each dye is short, there is no major bias coming from our modeling approach, although we do acknowledge the presence of two mutually compensatory small biases.

Second, although there is clearly at most a very small bias present in the practical case of our data, we also wanted to thoroughly consider the “ideal” case or thought excercise considered by the reviewer. As the reviewer suggested, we performed extensive Monte Carlo simulations following the Euler-Maruyama scheme to assess if there could be any bias coming from free molecules being over-counted due to continuous re-appearance. For simplicity, we will continue with the numbers provided by the reviewer (e.g. 50% bound) and for unspecified numbers use what we find for CTCF in mESCs. Thus, we will assume the following:

50% of molecules are bound and 50% are in free 3D diffusion.The free diffusion constant is the same as for Halo-mCTCF: 2.5 µm2/s.For simplicity, we assume the nucleus is approximately a cube with a side length of 8 µm. This assumption gives us an mESC nuclear volume very similar to that calculated for an mES nucleus as an ellipsoid (Chen et al., 2014). For purposes of our calculations, a cuboid nucleus greatly simplifies our simulations. The particles have to remain within this cube.We assume the laser excitation beam thickness is 4 µm under highly inclined and laminated optical sheet illumination which we used for single-molecule tracking. This number is based on Tokunaga et al., 2008. Thus, since the nucleus is modeled as a cube with a diameter of 8 µm, half the nucleus is being illuminated. For simplicity, we model the excitation beam as a step-function with uniform photobleaching probability.We assume that molecules photobleach with a first-order rate constant of λ.We will use our experimentally determined axial detection volume of 700 nm (so just below 1/10 of the nucleus like the reviewer suggested).The 2D localization error is 35 nm as in our experiments. Accordingly, the 1D localization error is 25 nm.The time between frames, Δτ, is 4.5 ms as in our experiments.Since the rate of unbinding (~1 min) and the rate of re-binding (~1 min) for CTCF is negligible at a 4.5 ms frame rate, we assume molecules never shift between the free and bound states (in practice, that they photobleach before they can shift).

We then simulate Brownian diffusion for 500,000 trajectories in each of the 3 dimensions by picking random numbers in each dimension from a normal distribution: N~(0, σ), where the mean is zero and σ=2DΔτ. The plots in Figure 7 show example 3D trajectories. We convert the part of all the 3D trajectories inside the detection volume into 2D trajectories and then we apply our SMT modeling approach to infer *D*_FREE_ and the fraction bound. The mean trajectory length was 3 frames similar to our experimental data. As can be seen, there is actually very low bias and using the entire trajectory (50.5% bound) or just the first 7 jumps (47% bound) give very similar results:

Author response image 3.**DOI:**
http://dx.doi.org/10.7554/eLife.25776.026

100 randomly chosen trajectories are shown on the right. Note that the trajectories outside the HiLo-excitation range are quite long because in this region there is no photobleaching. Conversely, in the second plot, in which only trajectories that spend at least one frame in the axial detection volume (3.65 µm to 4.35 µm) are shown, most trajectories are quite short (mean: 3 frames) because of the high rate of photobleaching. Thus, at high photobleaching rates like in our experimental data, these Monte Carlo simulations confirm that there is minimal bias from our SMT modeling approach in the estimation of both the fraction bound and the free diffusion constant.

Nevertheless, these simulations suggest that using the entire trajectory yields a smaller error (+0.5%) than using the first 7 jumps only (-3%). However, these simulations neglect another substantial bias against moving molecules. Experimentally, the more the molecules move, the more their photons are spread out, even under 1 ms stroboscopic illumination. Thus, the most mobile molecules have a lower signal-to-noise and fit less well to the 2-dimensional Gaussian PSF function that we use in our localization algorithm. The bound molecules conversely do not move and generally have a better signal-to-noise. It is very challenging to estimate how big the bias is from this, since it is difficult to model all experimental noise sources, but we expect it to be at least a couple of percentage points. Thus, although the approach suggested by the reviewer works slightly better for simulated data, because of this large bias against free molecules in the real data, we prefer to use the first 7 jumps only in our analysis of the experimental data. We also note that we included paSMT and modeling on Halo-3xNLS (overwhelmingly free) and H2b-Halo (overwhelmingly bound) as controls in Figure 3—figure supplement 1. These show the expected distributions (overwhelmingly free and overwhelmingly bound, respectively) and thus help verify, with orthogonal experimental controls, that there is no major bias coming from our approach.

Finally, we considered an ideal case with extremely photostable dyes (mean trajectory length: 100 frames). We note that this case is totally unrealistic given currently available dyes and microscope modalities and that we are about 2 orders of magnitude from this case even with state-of-the-art Janelia Fluor dyes today. In this experimentally unrealistic case, the reviewer is correct and our SMT modeling approach leads to a very serious undercounting of bound molecules (estimated fraction bound: 5%; actual fraction bound: 50%). As is also clear from the plots in Figure 8, the trajectories that appear in the axial detection volume are now extremely long.

Author response image 4.**DOI:**
http://dx.doi.org/10.7554/eLife.25776.027

In summary, while there is clearly no significant bias from our SMT modeling approach using our experimental data, we are working on a full generalization of our modeling approach to include dyes of variable stability and we agree that this is an important future direction, since dyes are improving quite rapidly. However, since this is not relevant to the present study, we will leave this for a future technical report. We have updated our discussion of the SMT modeling approach in the Materials and methods to include this important caveat and make clear that our modeling approach applies only to dyes with significant photobleaching.

*Furthermore, the estimation of the bound fraction and the estimation of the residence times are performed on two very different timescales (4.5-30 ms vs. >3s). It would be nice if the authors could exclude the presence of a fast dissociating population (in the gap between the two measurements), that might affect the calculation of the search times. Maybe by showing that the bound fraction does not change when acquiring data at a slower frame time (in the 50-200 ms range)?*

There are three populations: freely diffusing molecules; non-specifically bound CTCFs (fast dissociation) and specifically bound CTCFs (slow dissociation). Ideally, we would be able to image all three populations in the same experiment, but unfortunately this is not possible. In the fast-tracking experiments (Figure 3), we distinguish between freely diffusing molecules and the total bound fraction. In the slow-tracking experiments (Figure 2), we image only the bound population and distinguish between specifically bound population (slow component of the double-exponential) and the non-specifically bound population (fast component of the double-exponential). To see the stably bound population we need to use slow frame rates and low laser intensity (the fastest we imaged at was 300 ms; see Figure 2—figure supplement 1). In the 50-200 ms range which the reviewer suggested, it is not possible to capture the freely-diffusing population. To illustrate why, we did Monte Carlo simulations. We populate a 700 nm thick axial slice and randomly distribute molecules along this slice. We then simulate Brownian diffusion and calculate the fraction of molecules that remain within the focal plane for a biologically plausible range of diffusion constants (D = 1-12 µm2/s). The results are shown in Figure 9:

Author response image 5.**DOI:**
http://dx.doi.org/10.7554/eLife.25776.028

The dots correspond to simulation results and the lines correspond to a functional fit (similar to our Z-correction term). We allowed for 1 gap in the tracking and used a frame rate of 50 ms.

As can be seen, even for a slow-diffusing protein like CTCF with D ~ 2.5 µm^2^/s, most molecules leave the focal plane very quickly which leads to a huge undercounting of freely diffusing molecules. Thus, at the slower frame times of 50-200 ms there is too large a bias against freely diffusing molecules to allow accurate quantification. Moreover, tracking becomes extremely challenging, since molecules can easily travel across the whole nucleus in the 50-200 ms time range.

Finally, we note that although estimating the bound fraction from FRAP experiments is less quantitative (since there is significant diffusion in-and-out of the bleach spot during the long time it takes to bleach a circular spot), our FRAP experiments are in pretty good agreement with our SMT estimates. Consider for example Figure 2—figure supplement 3, showing FRAP for mRad21-SNAPf. Assuming full photobleaching (but not the appearance thereof due to diffusion into the spot during the slow bleach), after the initial recovery due to non-specific binding, there is ~50% bound molecules in S/G2 and ~40% in G1, which is in quantitative agreement with our SMT measurements (Table 1).

3) FRAP and modelling of FRAP.

*The details of the choice of the model for FRAP data are lacking. A double exponential assumes that diffusion is much faster than the unbinding process and the acquisition rate, and therefore does not contribute to recovery. This assumption needs to be tested/justified.*

*While it seems reasonable that most of the molecules will have diffused out from the bleach region in the time between two acquisitions (75%-80% given the diffusion coefficients reported by the authors), diffusion might still contribute to the first time-points of the recovery curve. The authors might test for the role of diffusion by plotting the evolution of the bleach profile over time, which should not broaden, if the assumption of the negligible role of diffusion is correct. Also, the estimations for CTCF residence time are quite different for SMT and FRAP (1 min vs. 3-4 min): the authors indicate that this might be due to multiple binding and unbinding events during the recovery. I have some doubts that in this case a multi-exponent fitting of the recovery curve is the most appropriate model. More importantly this interpretation is unlikely as the SMT data seem to indicate that the molecules spend about 60s between two stable binding events, and of these about 30s are spent diffusing freely in the nucleus. In these 30s the molecule should diffuse far away from the bleaching region, so it is unclear how the multiple binding events could significantly contribute to the recovery. One possibility is anomalous diffusion, that would constrain the spread of CTCF molecules after unbinding. Do the authors have any evidence from the SMT data for anomalous behavior of free CTCF molecules? Could relabelling by free dye during recovery [as reported by others] impact on the interpretation of the FRAP?*

We agree that our FRAP modeling section in the Materials and methods was too brief and we have now completely re-written the section and significantly extended it. We now motivate our choice of FRAP model based on the theoretical work by Sprague et al., 2004, that devised rules for the appropriate choice of FRAP model based on where one is in parameter space (Sprague et al., 2004). As is clear from both the reviewer’s calculations and Figure 3 in Sprague et al. we are in the “reaction dominant” parameter regime (kON*w2DFREE≪1, where w is the radius of the circular bleach spot and kOFFkON*≲1) and can thus ignore diffusion. In our quantification of FRAP recovery we consider a circle of radius 0.6 µm and as is clear from Figure 2—figure supplement 3 showing the radial bleach spot profile for Halo-mCTCF, mRad21-Halo and H2b-Halo this area is not significantly affected by diffusion (diffusion would cause recovery from the edges first, whereas binding will lead to mostly radially uniform recovery). For reference, compare these radial bleach profiles to Figure 2 in Mueller et al. (Mueller, Wach and McNally, 2008). As is clear, the examples in Mueller et al. show a case where diffusion cannot be ignored, but the flat radial profiles below 0.6 µm even for the first frame after the bleach (0 s; dark blue) in our case demonstrates that diffusion does not affect our FRAP recoveries.

Regarding “multiple binding and unbinding events”, we are sorry if the main text was unclear. We were not referring to the discrepancy. What we meant is that 1 residence time (RT) is not sufficient for full FRAP recovery, but that multiple binding and unbinding events are necessary for full FRAP recovery (we have noticed that some people confuse the RT with the time for full FRAP recovery). But we agree that our phrasing was poor and we have changed it in the main text.

Regarding anomalous diffusion, the reviewer is indeed correct that both CTCF and Rad21 exhibit anomalous diffusion. We suspect that clustering combined with anomalous diffusion may account for the discrepancy between our SMT RT estimate (1 min) and our FRAP estimates (3-4 min). But since we cannot say for sure at this stage, we prefer not to speculate too much in the main text. Moreover, previous studies have demonstrated that slight changes in the FRAP protocol and modeling assumptions (Mueller et al., 2008; Mazza et al., 2012) can significantly affect the RT estimate. Thus, we view FRAP-inferred RTs as rough estimates.

Finally, the reviewer points out that re-labeling by free dye could contribute to artifactual recovery. In fact, while this paper was under review at *eLife*, James Rhodes and Kim Nasmyth (Rhodes and Nasmyth, personal communication) also contacted us to inform us that they see re-labeling by free dye affecting their FRAP. They suggested a simple control experiment to test for this: add an excess of “dark” SNAP or Halo ligand to outcompete any re-labelling by low amounts of free dye that were not fully washed out. To test if this would have affected our FRAP studies, we repeated our FRAP experiments and compared FRAP recoveries of mESCs expressing H2B-SNAPf labeled with either 500 nM cp-SNAP-JF549 only followed by washings (red) or 500 nM cp-SNAP-JF549 followed by washings and then re-labelled with 500 nM “dark” SNAP-block (NEB #S9106S) (black), which remains in excess at 500 nM during the live-cell FRAP experiment. As is clear from Figure 2—figure supplement 3, there was no effect on our FRAP studies from re-labeling.

This is perhaps not surprising when you consider the timescales. In 11 min (our FRAP duration), ~1% new protein should be synthesized (11 min / 15 h; cell cycle time ~ 15 h). Thus, even if all newly synthesized proteins were immediately re-labeled, at most, this would contribute a 1% effect and due to cell drift and experimental noise, our total experimental error sources exceed 1% and thus we would not be able to detect such as small effect.

In conclusion, we acknowledge that our FRAP section in the Materials and methods was too brief and sparse and we thank the reviewers for pointing this out. We believe the current version comprehensively addresses all the issues raised by the reviewers.

*4) Two color dSTORM analysis*

(Subsection “CTCF and cohesin co-localize in cells and show a clustered nuclear organization”, Figure 4 and Figure 4—figure supplement 1).

*The two-color dSTORM image (Figure 4) shows more Rad21 (localization) signals than CTCF signals. This seems not in line with the ChIP-seq. peak numbers (68.000 CTCF, 33.400 Rad21, 97% overlap of Rad21 peaks with CTCF) that would lead to expect this to be the other way round. The authors should offer an explanation for this apparent inconsistency.*

When assessing ChIP-seq it is important to keep in mind the very different binding modes of CTCF and cohesin and that the ChIP-assay strongly favors CTCF over cohesin. CTCF is a sequence-specific binding protein with 11 zinc-finger domains, which is an unusually large DNA binding domain (most transcription factors have 2-4 zinc-fingers). Consequently, CTCF binds DNA directly at specific sequences with an unusually large DNA binding protein surface, thus providing an extensive protein surface to directly formaldehyde-crosslink to the DNA, which is necessary for ChIP-seq enrichment. Cohesin on the other hand is a large multiprotein complex (~50 nm) that topologically embraces DNA. Thus, there are likely no direct protein-DNA contacts in the case of cohesin. Moreover, the vast majority of the cohesin ring is composed of Smc1 and Smc3 – only a small fraction of the ring is made up of Rad21. Thus, protein-DNA crosslinking is almost certainly much, much more efficient for CTCF than for Rad21. More importantly, if the extrusion model holds, the vast majority of DNA-associated cohesin molecules are extruding along DNA at most times at quite high rates (perhaps even ~40-50 kb/min according to Wang, Brandao, Le, Laub, Rudner, Science, 2017). Thus, cohesin is largely spread out along the chromosomal DNA instead of being enriched at specific locations. ChIP-seq peaks are only called for regions where there is a high enrichment at a short sequence. Thus, it is perfectly possible for there to be many more DNA-associated cohesin molecules, yet fewer cohesin ChIP-seq peaks than CTCF peaks and we believe this to be the case.

*Furthermore, there seems to be a prominent enrichment of Rad21 along the nuclear envelope, which I have not been aware from other paper describing the nuclear distribution of cohesin using conventional imaging. Hence it may be useful to compare the dSTORM images with widefield deconvolution or confocal images of the same cell line(s).*

We thank the reviewer for spotting this. The reviewer is absolutely correct that Rad21 should not be enriched at the nuclear envelope and the appearance of this is a dye-labeling artifact. Whereas the labeling of HaloTag is extremely specific and essentially without any background, labeling of the SNAP-Tag is somewhat less specific. To demonstrate this, we performed an extensive series of dye-labeling and flow cytometry experiments that are shown in the new Figure 1—figure supplement 1. While using cp-JF549 instead of SNAP-TMR sold by NEB does substantially increase specificity (Figure 1—figure supplement 1 vs. 1C), there is still some artefactual labeling of the nuclear envelope for the SNAP-dyes (we know this from labeling cells that do not express any SNAP-tag proteins). This does not affect any of our results (e.g. when we do FRAP on mRad21-SNAPf, the bleach spot is never at the nuclear envelope), but it does lead to this confusing impression from Figure 4. We thank the reviewer for pointing this out and we have re-plotted Figure 4 from the raw data (list of single-molecule localizations) but excluded the nuclear envelope to avoid giving this false impression. We now state explicitly that there is this dye-labeling artifact in the dSTORM Materials and methods to make this limitation clear.

*For the additional clustering analyses (Figure 4—figure supplement 1) the authors use a 2D PALM localization approach. With the depth of the optical slice being around 700 nm, could this have potential effects on the clustering analyses? How should the clustering be interpreted? Do the clusters coincide with the co-localizing regions in the dSTORM images? Are the clusters equivalent to the LMCs, whereas non-clustering molecules are rather belonging to the free fraction? If 25% of CTCF or 35% of Rad21 molecules are in clusters, then only a subset of LMCs are in clusters, right? Then again, if 15% NLS-Halo is found in clusters, how relevant is the slight increase cluster fraction for CTCF and Rad21?*

The reviewers raise several important points here regarding what the functional roles of clusters are and if there could be technical artifacts.

Regarding the function of the clusters: given that there is an intriguing overlap of CTCF and cohesin clusters, it is very tempting to speculate that some of these clusters are composed of LMCs (we now state explicitly that the clusters overlap in the dSTORM images). However, it is technically extremely challenging to demonstrate unambiguously that the clusters hold together loops – we don’t know of any convincing way of showing this with currently available methods. Thus, we deliberately avoided making strong claims regarding the function of the clusters. We just report their existence and that they tend to be small (~30-40 nm radius). Regarding the biological relevance of there being 15-35% of molecules in clusters, we suspect that this is likely to be functionally important, but without being able to assign a clear function to the clusters, this is difficult to say at this stage. We nevertheless appreciate the questions the reviewers raise regarding their function and we hope to follow up on this in future studies, but those studies are beyond the scope of this manuscript.

Regarding the technical questions: we are confident in the existence of the clusters and we have gone well above-and-beyond the standards of the PALM/STORM field to confirm their existence. First, we use start-of-the-art and statistically rigorous Bayesian approaches to call clusters, instead of simple thresholding approaches which are difficult to assess in a statistically rigorous manner (Figure 4—figure supplement 1). Second, we show that most clusters are not a photoblinking artifact – we label the same cell with two spectrally distinct dyes and observe that most clusters contain close to a binomial distribution of dyes (Figure 4—figure supplement 1). To our knowledge, this is the strongest possible evidence for clustering available and we can think of no other way of explaining Figure 4—figure supplement 1 other than the existence of clusters. Third, we use “negative controls”. Since the single-molecule nuclear organization field is still in its infancy, there is not a well-known protein with a truly random nuclear organization. In lieu of this, we use Halo-3xNLS as a negative control. The reviewer is correct that we see some clustering for Halo-3xNLS (~15%), but as is evident from Figure 4—figure supplement 1, the error bars are small and the increase in clustering for CTCF and Rad21 is highly statistically significant.

Nevertheless, we acknowledge two limitations of the current approach. First, our current imaging is in 2D. In the future, we plan to follow up with 3D-PALM using adaptive optics to achieve a sufficient axial resolution. Second, our 2-color dSTORM was performed in unsynchronized interphase cells. In the future, we will follow up with cell-cycle synchronized experiments to avoid potentially confounding cohesin clusters coming from molecules mediating sister-chromatid cohesion.

Thus, in summary, we deliberately refrained from making strong claims about clustering although we have shown that CTCF and cohesin form clusters using two different imaging modalities (PALM and dSTORM), and we have performed two different control experiments to demonstrate that they are not an artifact (Halo-3xNLS negative control and 2-color dye experiment) and we have used the most statistically rigorous available approaches to calling clusters (Figure 4—figure supplement 1).

Subsection “CTCF and cohesin exhibit distinct nuclear search mechanisms” and Table 1: In the text 3 fractions are distinguished (bound to cognate site; non-specifically associated with chromatin; free 3D diffusion), whereas in the table only two fractions are distinguished (bound total vs. bound specific). Why not listing the 3 fractions as in the text, to avoid confusion? Could the "non-specific binding" or "1D sliding" not also explained by corralled or anomalous diffusion? Does the Rad21-HaloTag kinetic data imply that Rad21 (and other Cohesin subunits) are predominantly present as either free (searching) or bound complexes, but not as single proteins?

We thank the reviewer for this suggestion – we agree that listing the three fractions (specifically bound, non-specifically bound, free 3D diffusion) separately is simpler and have changed Table 1 accordingly. We do see anomalous diffusion and plan to explore this further in future studies – however, the anomalous diffusion we see appears to hold on all spatiotemporal scales (suggesting perhaps a compact exploration mechanism – we will explore this further in the future) and thus, the anomalous diffusion holds for displacements on the hundreds of nm scale (and thus will be counted in the free 3D diffusion fraction). “1D sliding” and “non-specific binding” would appear as displacements below our localization error and thus would not be counted in the free 3D fraction. Therefore, the anomalous diffusion that we do observe cannot explain the 1D sliding. However, we do not know what the exact nature of the “non-specific binding” is and it is by its very nature extremely difficult to study inside cells. We therefore prefer not to make strong statements about it.

Finally, our SMT data does in fact indicate that essentially all mRad21-Halo is incorporated into the cohesin complex – we know this because, when we consider the Rad21 mutant (F601R, L605R, Q617K; Figure 3) that cannot form cohesin complexes or even when we over-express wt Rad21 (Figure 3—figure supplement 1), we see much faster diffusion. This suggests that “monomeric” Rad21-Halo has a much faster diffusion constant (~4 µm^2^/s) than Rad21-Halo incorporated into cohesin complexes (~1.5 µm^2^/s). We thank the reviewer for pointing this out – it is an important point – and we now state this in the manuscript.

[Editors' note: further revisions were requested prior to acceptance, as described below.]

*Reviewer #3:*

*In the revised version of the manuscript, Hansen et al. have made substantial efforts to answer the questions raised, by performing computer simulations and restating parts of the text. I am a little disappointed that despite some additional "wet" experiments were requested, the authors did not perform any. Nevertheless their arguments against my criticism are now more convincing, and I support publication in eLife.*

We thank the reviewer for supporting publication in *eLife*. We would like to clarify, that we actually did perform new “wet” experiments during the revision: Figure 1—figure supplement 1 is a completely new figure based on dye labeling and flow cytometry experiments we performed to address parts of the reviewer’s important observation of non-specific nuclear membrane staining by SNAP dyes in Figure 4 of our original submission. Moreover, Figure 2—figure supplement 3 shows new FRAP experiments that we did to address the reviewer’s concern of “re-binding” of unwashed dye during FRAP imaging, which was also recently reported on BioRxiv (Rhodes & Nasmyth, BioRxiv: http://www.biorxiv.org/content/early/2017/04/04/124107; we now cite this preprint in the methods). We believe these experiments have strengthened the manuscript.

*The authors performed computer simulations to prove that limiting the population of the distribution of displacements to the first 7 frames of each track, however I see some problems with these simulations.*

*For the paper purpose, my suggestion would be to drop the description of these simulations from the text: they have a better argument, given by the fact that whether they analyze their experimental data by looking at all of the jumps, or only at the first 7 they essentially obtain the same results.*

Although the reviewer’s logic and calculations are correct, their perceived problem with the simulations is based on an incorrect assumption. Thus, we maintain that our simulations are correct. In fact, we very much appreciate the reviewer pushing us to do these simulations to further validate our approach. Consequently, the paSMT modeling has now been validated by both simulations and biological controls (such as H2B-Halo and Halo-3xNLS) within the context of our experimental regime of photobleaching. Thus, since the simulations are correct and confirm our approach, we would like to keep them in the methods as this seems to be the most straightforward and comprehensive approach that also strengthens the manuscript.